# CONFOUNDER IDENTIFICATION-FREE CAUSAL VISUAL FEATURE LEARNING

## ABSTRACT

Confounders in deep learning are in general detrimental to model's generalization where they infiltrate feature representations. Therefore, learning causal features that are free of interference from confounders is important. Most previous causal learning-based approaches employ back-door criterion to mitigate the adverse effect of certain specific confounders, which require the explicit identification of confounders. However, in real scenarios, confounders are typically diverse and difficult to be identified. In this paper, we propose a novel **C**onfounder **I**dentification-free **C**ausal Visual **F**eature Learning (*CICF*) method, which obviates the need for identifying confounders. *CICF* models the interventions among different samples based on the front-door criterion, and then approximates the global-scope intervening effect based on the instance-level intervention from the perspective of optimization. In this way, we aim to find a reliable optimization direction, which eliminates the confounding effects of confounders, to learn causal features. Furthermore, we uncover the relation between *CICF* and the popular meta-learning strategy MAML (Finn et al., 2017), and provide an interpretation of why MAML works from the theoretical perspective of causal learning for the first time. Thanks to the effective learning of causal features, our *CICF* enables models to have superior generalization capability. Extensive experiments on domain generalization benchmark datasets demonstrate the effectiveness of our *CICF*, which achieves the state-of-the-art performance.

## 1 INTRODUCTION

Deep learning excels at capturing correlations between the inputs and labels in a data-driven manner, which has achieved remarkable successes on various tasks, such as image classification, object detection, and question answering (Liu et al., 2021; He et al., 2016; Redmon et al., 2016; He et al., 2017; Antol et al., 2015). Even so, in the field of statistics, *correlation is in fact not equivalent to causation* (Pearl et al., 2016). For example, when tree branches usually appear together with birds in the training data, deep neural networks (DNNs) are easy to mistake features of tree branches as the features of birds. A close association between two variables does not imply that one of them causes the other. Capturing/modeling correlations instead of causation is at high risk of allowing various confounders to infiltrate into the learned feature representations. When affected by intervening effects of confounders, a network may still make correct predictions when the testing and training data follow the same distribution, but fails when the testing data is out of distribution. This harms the generalization capability of learned feature representations. Thus, learning causal feature, where the interference of confounders is excluded, is important for achieving reliable results.

As shown in Fig. 1, confounders $C$ bring a spurious (non-causal) connection $X \leftarrow C \rightarrow Y$ between samples $X$ and their corresponding labels $Y$. A classical example to shed light on this is that we can instantiate $X, Y, C$ as the sales volume of ice cream, violent crime and hot weather. Seemingly, an increase in ice cream sales $X$ is correlated with an increase in violent crime $Y$. However, the hot weather is the common cause of them, which makes an increase in ice cream sales to be a misleading factor of analyzing violent crime. Analogically, in deep learning, once the misleading features/confounders are captured, the introduced biases may be mistakenly fitted by neural networks, thus leading to the detriment of the generalization capability of learned features. In theory, we expect DNNs to model the causation between $X$ and $Y$. Deviating from such expectation, the interventions of confounders $C$ make the learned model implicitly condition on $C$. This makes that

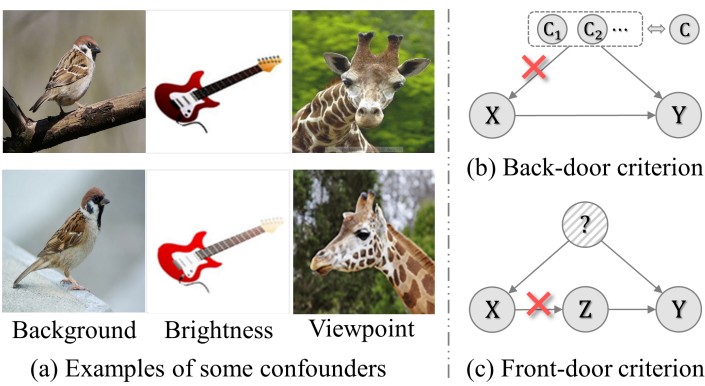

Background   Brightness   Viewpoint

(a) Examples of some confounders

(b) Back-door criterion

(c) Front-door criterion

Figure 1: (a) Examples of some confounders, which may lead to learning biased features. (b) Back-door criterion in causal inference, where the counfunders are accessible. (c) Front-door criterion in causal inference, where the confounders are inaccessible.

the regular feature learning does not approach the causal feature learning. To learn causal features, previous studies (Yue et al., 2020; Zhang et al., 2020; Wang et al., 2020b) adopt the backdoor criterion (Pearl et al., 2016) to explicitly identify confounders that should be adjusted for modeling intervening effects. However, they can only exploit the confounders that are accessible and can be estimated, leaving others still intervening the causation learning. Moreover, in many scenarios, confounders are unidentifiable or their distributions are hard to model (Pearl et al., 2016).

Theoretically, front-door criterion(Pearl et al., 2016) does not require identifying/explicitly modeling confounders. It introduces an intermediate variable $Z$ and transfers the requirement of modeling the intervening effects of confounders $C$ on $X \rightarrow Y$ to modeling the intervening effects of $X$ on $Z \rightarrow Y$. Without requiring explicitly modeling confounders, the front-door criterion is inherently suitable for wider scenarios. However, how to exploit the front-door criterion for causal visual feature learning is still under-explored.

In this paper, we design a Confounder Identification-free Causal visual Feature learning method (*CICF*). Particularly, *CICF* models the interventions among different samples based on the front-door criterion, and then approximates the global-scope intervening effect based on the instance-level interventions from the perspective of optimization. In this way, we aim to find a reliable optimization direction, which eliminates the confounding effects of confounders, to learn causal features. There are two challenges we will address for *CICF*. 1) How to model the intervening effects from other samples on a given sample in the training process. 2) How to estimate the global-scope intervening effect across all samples in the training set to find a suitable optimization direction.

As we know, during training, each sample intervenes others through its effects on network parameters by means of gradient updating. Inspired by this, we propose a gradient-based method to model the intervening effects on a sample from all samples to learn causal visual features. However, it is intractable to involve such modeled global-scope intervening effects in the network optimization, which requires a traversal over the entire training set and is costly. To address this, we propose an efficient cluster-then-sample algorithm to approximate the global-scope intervening effects for feasible optimization. Moreover, we revisit the popular meta-learning method Model-Agnostic Meta-Learning (MAML) (Finn et al., 2017). We surprisingly found that our *CICF* can provide an interpretation on why MAML works well from the perspective of causal learning: MAML tends to learn causal features. We validate the effectiveness of our *CICF* on the Domain Generalization (DG) (Wang et al., 2021; Zhou et al., 2021a) task and conduct extensive experiments on the PACS, Digits-DG, Office-Home, and VLCS datasets. Our method achieves the state-of-the-art performance.

## 2 RELATED WORK

**Causal Inference** aims at pursuing the causal effect of a particular phenomenon by removing the interventions from the confounders (Pearl et al., 2016). Despite its success in economics (Rubin, 1986), statistics (Rubin, 1986; Imbens & Rubin, 2015) and social science (Murnane & Willett, 2010), big challenges present when it meets machine learning, *i.e.*, how to model the intervention

from the confounders and how to establish the causal model. A growing number of works have moved a step forward by taking advantage of the back-door criterion (Pearl et al., 2016) on various tasks, *e.g.*, few-shot classification (Yue et al., 2020), vision-language task (Wang et al., 2020b), domain adaptation (Yue et al., 2021), class-incremental learning (Hu et al., 2021), and semantic segmentation (Zhang et al., 2020). Limited by the back-door criterion, most of them are required to identify and model the distributions of the confounders. However, this may be challenging in the real world because confounders are typically diverse and usually appear implicitly. To get rid of the dependency on confounders, Yang et al. (2021), for the first time, propose to utilize the front-door criterion to establish a causal attention module for vision-language task. However, it still requires the modeling of intervention in the testing stage, which is complicated.

In contrast, this work is the first attempt to apply the front-door criterion for learning causal visual features by considering the intervention among samples. Ours improves the generalization ability of DNNs from the optimization perspective and is confounder identification-free.

**Model Generalization** plays a prominent role for DNNs to be applied in real-world scenarios. To improve the performance on the testing dataset which has distribution shift (Sun et al., 2016) with training data, various domain generalization (DG) (Muandet et al., 2013; Zhou et al., 2021a; Wang et al., 2021; Shen et al., 2021; Wei et al., 2021) methods have been proposed. In general, these methods can be divided into three categories, *i.e.*, domain-invariant representation learning, data or feature manipulation, and meta-learning. The first category intends to learn domain-invariant features that follow the same distributions (Muandet et al., 2013; Li et al., 2018b; Taori et al., 2020; Li et al., 2018d; Motiian et al., 2017; Mahajan et al., 2021b; Jin et al., 2020). The second category aims to improve the generalization ability of models through enriching the diversity of source domains, either in image space (*e.g.*, CrossGrad (Shankar et al., 2018), DDAIG (Zhou et al., 2020a) and M-ADA (Qiao et al., 2020)), or feature space (MixStyle (Zhou et al., 2021c) and RSC (Huang et al., 2020)). Another new line of DGs utilize meta-learning as training strategy (Zhao et al., 2021; Liu et al., 2020; Li et al., 2018a; 2020; Wei et al., 2021; Balaji et al., 2018). MAML (Finn et al., 2017) takes the advantage of meta-learning to find a good parameters initialization for fast adaptation to new tasks. Following MAML, Li et al. (2018a); Dou et al. (2019); Li et al. (2020); Balaji et al. (2018) introduce meta-learning into DG to simulate domain shift or learn domain-invariant parameters regularizer during training. Other variants of DGs exploit episodic training (Li et al., 2019) and ensemble learning (Zhou et al., 2021b; Seo et al., 2020; Cha et al., 2021).

In this paper, from a new perspective, we propose a scheme for model generalization termed as Confounder Identification-free Causal Visual Feature Learning (*CICF*).

## 3 PROPOSED METHOD

In this section, we first depict a supervised learning process in a causal graph (Pearl, 2009b), and uncover the stumbling effects of confounders which prevent the achievement of high generalization capability of models in Sec. 3.1. Then, in Sec. 3.2, based on the front-door criterion, we elaborate our Confounder Identification-free Causal Visual Feature Learning(*CICF*) from two perspectives, respectively as how to model mutual intervening effects between different instances and how to approximate such intervening effects from the global scope. Furthermore, we describe our *CICF* which alleviates the intervening effects from the optimization perspective in Sec. 3.2. In Sec. 3.3, we uncover the relation between our *CICF* and the popular meta-learning strategy MAML (Finn et al., 2017), and provide an interpretation of why MAML works from the theoretical perspective of causal inference.

### 3.1 PROBLEM DEFINITION AND ANALYSIS

Given a training dataset with input and label pairs $\{X, Y\}$, the goal of training Deep Neural Networks is to learn/capture the causation between input samples $X$ and prediction labels $Y$, *i.e.*, the conditional probability $P(Y|do(X))$. As shown in Fig. 2 (a), we parameterize the network as $\varphi$ and separate it into two successive parts, *i.e.*, $h$ and $f$.

DNNs capture label-associated features which are not necessarily the casual ones due to the intervening effects of confounders, such as background, brightness, and viewpoint. We denote the intermediate features and confounders as $Z$ and $C$, respectively. Fig. 2 illustrates the relations in

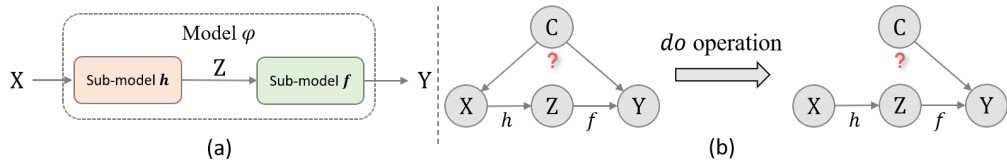

Figure 2: Causal graph in supervised learning. (a) A network/model $\varphi$ consists of sub-models $h$ and $f$. $X$ and $Y$ denote the input and prediction labels, respectively. (b) Causal graph where *do* operation aims to remove the intervening from confounder $C$.

a causal graph. Intervened by the confounders, the conditional probability $P(Y|X)$ learned by the model $\varphi$ actually involves two paths, *i.e.*, $X \to Z \to Y$ and $X \leftarrow C \to Y$. $X \to Z \to Y$ denotes the expected causal effect from the input samples $X$ to their prediction labels $Y$. The path $X \leftarrow C \to Y$ denotes the non-causal correlation between $X$ and $Y$ due to their common cause $C$, which may introduce biases into the learning of $P(Y|X)$ and thus affect the generalization capability of model $\varphi$.

Intuitively, it is crucial to get rid of the harmful bias from those confounders for causal feature learning. In the literature of causal inference (Pearl et al., 2016), the confounding effects from confounders can be removed through the *do* operation (Pearl et al., 2016) by cutting off the connection from $C$ to $X$, as illustrated in Fig. 1 (b). With the definition of *do* operation, the real causation from $X$ to $Y$ can be formulated by $P(Y|do(X))$. The objective of our *CICF* is to learn features representation conforming to $P(Y|do(X))$.

**Back-door criterion.** In previous works (Yue et al., 2020; Wang et al., 2020b; Hu et al., 2021), when $C$ is identifiable, the back-door criterion (Pearl et al., 2016) is typically utilized to achieve the *do* operation as:

$$P(Y|do(X = x)) = \sum_c P(Y|X = x, C = c)P(C = c), \tag{1}$$

which acquires access to the distributions of all confounders $c \in C$. However, in many scenarios, the intervening effects are caused by unobservable or implicit factors. It is not feasible to identify the distribution of $C$ during training, limiting the usage of the back-door criterion.

**Front-door criterion.** For unidentifiable confounders, the **F**ront-door criterion (Pearl et al., 2016) provides us with a more practical alternative to **E**stimate the **I**ntervening **E**ffect, called *FEIE*, eschewing the identification of confounders $C$. Specifically, it introduces an intermediate variable $Z$ to help assess the effect of $X$ on $Y$, *i.e.*, $P(Y|do(X))$, which can be formulated as:

$$P(Y|do(X = x)) = \sum_z P(Z = z|X = x) \sum_{\tilde{x} \in X} P(Y|Z = z, \tilde{x})P(\tilde{x}), \tag{2}$$

where $Z = h(X)$, $\tilde{x} \in X$ denotes a sample from training data. Note that the effect of $X$ on $Z$ is identifiable because they have no common causes. In other words, there is no backdoor path from $X$ to $Z$. Thus, we have $P(Z = z|do(X = x)) = P(Z = z|X = x)$.

Front-door criterion is attractive for eliminating interventions. However, it is still under-explored for visual feature learning, where there is a lack of a simple and practical mechanism to exploit this theory for enhancing the generalization capability of models.

## 3.2 Confounder Identification-free Causal Visual Feature Learning

In this paper, we aim to achieve *Confounder Identification-free Causal Visual Feature Learning*, obviating the need for confounder identification. As indicated by Eq. (2), thanks to the front-door criterion, we do not need to identify and explicitly model confounders $C$. Despite this, it still imposes a challenge on how to accurately model the term $\sum_{\tilde{x}} P(Y|Z = z, \tilde{x})$ in the network training process. We treat the first part of $\varphi$ as the model $h$ to obtain the intermediate variable $z$, *i.e.*, $z = h(x)$. Because the parameters of $h$ are fixed in the inference stage and $z = h(x)$ is known

given any $x \in X$, $P(Z = z | X = x)$ is equal to 1 [1]. Thus, we can re-write Eq. (2) as:

$$P(Y|do(X = x)) = \sum_{\tilde{x} \in X} P(Y|Z = h(x), \tilde{x}) P(\tilde{x}), \tag{3}$$

where $P(Y|Z = h(x), \tilde{x})$ denotes the mutual intervening effects to the causation path $Z \to Y$ from another sample $\tilde{x}$. With the summation operation, Eq. 3 represents the *global* intervening effects accumulated from all samples in the training data. We will describe the instantiations of $P(Y|Z = h(x), \tilde{x})$ and the accumulation in Eq. (3) respectively as below.

**A Gradient-based Instantiation of *FEIE*.** Referring to the practices in prior works (Yue et al., 2020; Wang et al., 2020a) upon the back-door criterion, a straightforward method for modelling $P(Y|Z = h(x), \tilde{x})$ is to directly concatenate $z = h(x)$ and the feature of $\tilde{x} \in X$ before feeding them into $f$. However, this method would easily lead to a trivial solution once the information of $\tilde{x}$ is ignored by the layers of the neural networks. In contrast, we propose to explicitly model the intervening effects of $\tilde{x}$ on $Z \to Y$ with a gradient-based instantiation. We notice that, in the training process, *the influence of one instance on others can be reflected on the parameters updating with the gradient obtained based on this instance*. Therefore, for a given sample $x$, we propose to explicitly model the intervening effects $P(Y|Z = h(x), \tilde{x})$ of another sample $\tilde{x}$ on $x$ through $f$ as:

$$P(Y|Z = h(x), \tilde{x}) = f_{\theta_{\tilde{x}}}(Z = h(x)), \ where \ \theta_{\tilde{x}} = \theta - \alpha g_{\tilde{x}}, \ g_{\tilde{x}} = \nabla_{\theta} \mathcal{L}(f_{\theta}(h(\tilde{x}), \tilde{y})), \tag{4}$$

$f_{\theta}$ and $f_{\theta_{\tilde{x}}}$ denote the model $f$ before and after the parameters updating respectively, $g_{\tilde{x}}$ denotes the calculated gradient with respect to the sample $\tilde{x}$ and its label $\tilde{y}$. $\mathcal{L}$ and $\alpha$ represent the loss of cross entropy and learning rate, respectively. Incorporating Eq. (4) into Eq. (3), we have Eq. (5) as below to explicitly eliminate the interventions of all samples on the sample $x$ as:

$$P(Y|do(X = x)) = \sum_{\tilde{x} \in X} f_{\theta_{\tilde{x}}}(Z = h(x)) P(\tilde{x}). \tag{5}$$

**Global-scope Intervening Effects Approximation.** With the above introduced gradient-based instantiation, the globally accumulative intervening effects from all the training samples can be estimated by a traversal on $X$, which, however, is time- and memory-consuming in practice. To achieve an efficient estimation in the global scope, we apply the first-order Taylor's expansion on Eq. (5):

$$P(Y|do(X = x)) = \sum_{\tilde{x} \in X} [f_{\theta}(h(x)) - \alpha g_{\tilde{x}} \nabla_{\theta} f_{\theta}(h(x)) + o\left(\nabla_{\theta} f_{\theta}(h(x))\right)] P(\tilde{x})$$

$$\approx f_{\theta}(h(x)) - \alpha(\sum_{\tilde{x} \in X} g_{\tilde{x}} P(\tilde{x})) \nabla_{\theta} f_{\theta}(h(x)). \tag{6}$$

Eq. (6) reveals that the key to estimating $P(Y|do(X))$ lies in computing the global-scope gradient $g_{\dagger} = \sum_{\tilde{x} \in X} g_{\tilde{x}} P(\tilde{x})$ over all $\tilde{x} \in X$ accumulated via weighted sum with $P(\tilde{x})$ as the weight.

However, it is intractable to directly compute the global-scope gradient $g_{\dagger}$ by traversing over all the training samples. *Alternatively, we can traverse over a sampled small subset that shares the similar data distribution to that of all the training data.* As we know, when the training data are unbalanced and diverse, random sampling of a small subset would result in bias that mismatches the distribution of the dataset, leading to an inaccurate estimation of the global-scope gradient. To better estimate the data distribution and thus approach the global-scope gradient, we propose a sampling strategy dubbed as *clustering-then-sampling*. More discussion/analysis can be found in the **Appendix** A.1. Concretely, we first cluster the training samples of each class in the dataset into $K$ clusters with $K$-means algorithms (Pelleg et al., 2000) and totally obtain $K^{\dagger}$ clusters for the whole training data.

It is noteworthy that we found the samples in each cluster usually have similar gradient directions in optimization. Thus we represent each cluster with fewer samples randomly sampled from the same cluster, avoiding traversing over all the data. Then, the global-scope gradient $g_{\dagger}$ can be approximated with weighted sum over the sampled $M = \sum_{k=1}^{K^{\dagger}} N_k$ samples from $K^{\dagger}$ clusters:

$$g_{\dagger} = \frac{1}{M} \sum_{k=1}^{K^{\dagger}} \sum_{j=1}^{N_k} g_{\tilde{x}_{j,k} \in K_k}, \tag{7}$$

---

[1]We provide proof in Appendix A.3 that this satisfies the front-door criterion.

where $N_k$ is the number of instances sampled from the $k$-th cluster (being proportional to the size of this cluster), $g_{\tilde{x}_{j,k} \in K_k}$ denotes the gradients of the sample $\tilde{x}_{j,k} \in K_k$. Combined with Eq. (7), we rewrite Eq. (6) as:

$$P(Y|do(X = x)) \approx f_\theta(h(x)) - \alpha g_\dagger \nabla_\theta f_\theta(h(x)). \tag{8}$$

**Causal Visual Feature Learning.** *Based on the above theoretical analysis and the proposed intervention approximation strategy, the intractable causal conditional probability $P(Y|do(X))$ can be approximated based on Eq.(8), without requiring the identification of confounders $C$.* Actually, the Eq. (8) can be viewed as the first-order Taylor's expansion of $f_{\theta - \alpha g_\dagger}(h(x))$. Thus, we have:

$$P(Y|do(X = x)) \approx f_{\theta - \alpha g_\dagger}(h(x)), \tag{9}$$

here let $\theta_\dagger = \theta - \alpha g_\dagger$ (the parameters of $f$), which are updated with the global-scope gradient $g_\dagger$. The output of a model is thus denoted as $\hat{y}_{do(x)} = f_{\theta_\dagger}(h(x))$, which has been aware of the global-scope interventions from all other samples on the current sample $x$ based on such global-scope gradient updated model (*i.e.*, $f_{\theta_\dagger}$). Then, we can train an *unbiased* model $f$ to learn the causal visual features with the loss of cross-entropy:

$$\mathcal{L}_{CICF} = \sum_{x \in X} \mathcal{L}_{ce} \left( f_{\theta_\dagger}(h(x)), y \right), \tag{10}$$

where $y$ is the corresponding ground-truth label for $x$. The overall algorithm of *Confounder Identification-free Causal Visual Feature Learning* is described in Alg. 1 of Appendix.

Note that clustering-then-sampling is better than random sampling to approach the distribution of the training dataset, thereby being capable of approximating the global-scope gradient more accurately. We have theoretically analyzed that *clustering-then-sampling* has a more minor standard error (SE) for estimating the distribution of all the training data than random sampling, *i.e.*, $SE_{ours} < SE_{random}$ in the **Appendix** A.1. This demonstrates that our *clustering-then-sampling* is a more efficient and more accurate strategy to estimate the data distribution and then the global-scope gradient.

## 3.3 DISCUSSION

In this section, we will provide an analysis and comparison between our *CICF* and MAML (Finn et al., 2017). For the first time, we interpret why MAML works from a causal learning perspective, which is supported by our analysis in the previous subsections.

In the seminal work MAML (Finn et al., 2017), Finn *et al.* propose a model-agnostic meta-learning strategy that treats a batch of data as meta-train and another batch of data as meta-test for optimization. Particularly, given $T$ sets of data $\{D_{tr}^t, D_{te}^t\}_{t=1}^T$ corresponding to $T$ tasks $\{\mathcal{T}_t\}_{t=1}^T$, where $D_{tr}^t$ and $D_{te}^t$ denote meta-train and meta-test data respectively, the loss function of MAML for optimization can be represented as:

$$\mathcal{L}_{MAML} = \sum_t \mathcal{L} \left( f_{\theta_{tr}^t}(X_{te}^t), Y_{te}^t \right), \tag{11}$$

where $\theta_{tr}^t$ refers to the parameter virtually updated with the gradient $g_{tr}^t$ calculated on $D_{tr}^t$, *i.e.*, $\theta_{tr}^t \leftarrow \theta - \alpha g_{tr}^t$, which is treated as meta-train task. The optimization on $D_{te}^t$ is treated as meta-test task, where the parameters are updated as $\theta_{te}^t \leftarrow \theta - \alpha \nabla_\theta \mathcal{L}_{MAML}$. They interpret why MAML works from the perspective that it can provide a good parameter initialization which is robust for fast adaptation to new data. However, there is a lack of theoretical analysis and support in Finn et al. (2017). Based on our analysis in Section 3.2, for the first time, we have *a new understanding of the previous uses of MAML (Finn et al., 2017; Li et al., 2018a) (see Eq. (11)) from the perspective of causal inference.* $\mathcal{L} \left( f_{\theta_{tr}^t}(X_{te}^t), Y_{te}^t \right)$ in Eq. (11) actually models the intervention from meta-train data $D_{tr}^t$ to meta-test data $D_{te}^t$ within the task $t$ and endeavors to eliminate such local data modeled intervention. However, as revealed by our theoretical analysis in Section 3.2, learning reliable causal features requires the capturing and modeling of interventions from all the samples (*i.e.*, global interventions). There is no such solution in the previous works while we provide a practical and efficient one to model and eliminate the global-scope interventions in this paper. This enables reliable causal feature learning and promotes the achievement of higher generalization capability of models.

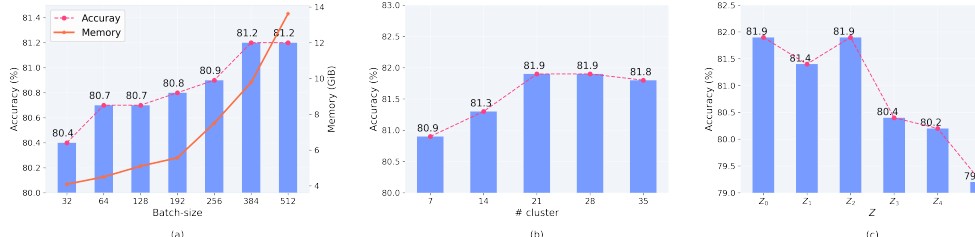

Figure 3: Ablation study on the influence from (a) batch-size, (b) the number of clusters, for the global-scope intervention estimation, and (c) different choices of $Z$ (c.f. Fig. 7). $\{Z_0, ..., Z_5\}$ denotes the features from the shallow layer to the deep layer. All results are averaged over four domains on PACS.

Table 1: Classification accuracy (%) of different DG methods on PACS with ResNet-18 and Digits-DG with a CNN backbone (Zhou et al., 2021c). ‡: our reimplemented results using the official code are different from the ones reported in the original paper.

| Method | PACS | | | | | Digits-DG | | | | |
|---|---|---|---|---|---|---|---|---|---|---|
| | A | C | P | S | Avg. | MINIST | MINIST-M | SVHN | SYN | Avg. |
| MMD-AAE | 75.2 | 72.7 | 96.0 | 64.2 | 77.0 | 96.5 | 58.4 | 65.0 | 78.4 | 74.6 |
| CCSA | 80.5 | 76.9 | 93.6 | 66.8 | 79.4 | 95.2 | 58.2 | 65.5 | 79.1 | 74.5 |
| JiGen | 79.4 | 75.3 | 96.0 | 71.6 | 80.5 | 96.5 | 61.4 | 63.7 | 74.0 | 73.9 |
| CrossGrad | 79.8 | 76.8 | 96.0 | 70.2 | 80.7 | 96.7 | 61.1 | 65.3 | 80.2 | 75.8 |
| MLDG | 79.5 | 77.3 | 94.3 | 71.5 | 80.7 | 94.7 | 60.3 | 61.5 | 75.4 | 72.6 |
| MASF | 80.3 | 77.2 | 95.0 | 71.7 | 81.1 | - | - | - | - | - |
| MetaReg | 83.7 | 77.2 | 95.5 | 70.3 | 81.7 | - | - | - | - | - |
| RSC | 83.4 | 80.3 | 96.0 | 80.9 | 85.2 | - | - | - | - | - |
| MatchDG | 81.3 | **80.7** | **96.5** | 79.7 | 84.6 | - | - | - | - | - |
| MixStyle‡ | 83.0 | 78.6 | 96.3 | 71.2 | 82.3 | 96.5 | 63.5 | 64.7 | 81.2 | 76.5 |
| FACT | 85.4 | 78.4 | 95.2 | 79.2 | 84.5 | **97.9** | 65.6 | 72.4 | **90.3** | 81.5 |
| ERM | 77.0 | 75.9 | 96.0 | 69.2 | 79.5 | 95.8 | 58.8 | 61.7 | 78.6 | 73.7 |
| ERM+MAML | 77.0 | 74.5 | 94.8 | 72.1 | 79.6 | 96.0 | 63.1 | 65.0 | 81.1 | 76.5 |
| ERM+*CICF* | 80.7 | 76.9 | 95.6 | 74.5 | 81.9 | 95.8 | 63.7 | 65.8 | 80.7 | 76.5 |
| ERM* | 82.5 | 74.2 | 95.4 | 76.5 | 82.1 | 96.1 | 65.0 | 73.0 | 84.6 | 79.7 |
| ERM*+MAML | 81.8 | 73.2 | 94.8 | 75.7 | 81.4 | 96.2 | 67.0 | 74.0 | 84.1 | 80.3 |
| ERM*+*CICF* | **84.2** | 78.8 | 95.1 | **83.2** | **85.3** | 95.6 | **68.8** | **76.5** | 86.0 | **81.7** |

## 4 EXPERIMENTS

To validate the effectiveness of our *CICF*, we apply it on the Domain Generalization (DG) (Zhou et al., 2021a; Wang et al., 2021) task, where the models are expected to learn the unbiased causal features to be generalized to different domains. We describe the datasets and implementation details in Sec. 4.1. Then we clarify the effectiveness of each component of our *CICF* in Sec. 4.2, and compare with previous methods in Sec. 4.3. Finally, we qualitatively show that *CICF* captures the causal features in Sec. 4.4.

### 4.1 DATASETS AND IMPLEMENTATION DETAILS

**Datasets.** We evaluate our method on four commonly used benchmark datasets (*i.e.*, PACS (Li et al., 2017), Digits-DG, Office-Home (Venkateswara et al., 2017) and VLCS (Torralba & Efros, 2011)) for Domain Generalization. 1) **PACS** (Li et al., 2017) contains images from four domains, *i.e.*, Photo (P), Art painting (A), Cartoon (C), and Sketch (S). Each domain consists of images in seven object categories. 2) **Digits-DG** is composed of four digit datasets, including MINIST (LeCun et al., 1998), MINIST-M (Ganin & Lempitsky, 2015), SVHN (Netzer et al., 2011) and SYN (Ganin & Lempitsky, 2015). Each dataset is regarded as a domain, which contains ten digit categories from zero to nine. 3) **Office-Home** is divided into four domains, including Artistic, Clipart, Product and Real World. There are 65 object categories related to the scenes of office and home. 4) **VLCS** (Torralba &

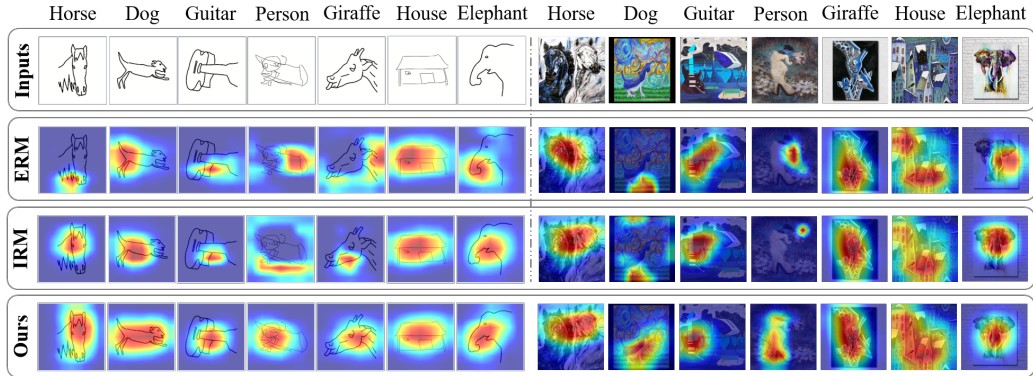

Figure 4: Visualization of the Grad-CAMs w.r.t. classification task for domain generalization. The first row of left/right panels shows the images from sketch/art painting (Art) domains in PACS. The second, third and fourth rows show the Grad-CAMs of ERM, IRM (Arjovsky et al., 2019) and our *CICF*.

Efros, 2011) covers images of five object categories from four domains, *i.e.*, PASCAL VOC 2007, LabelMe, Caltech, and Sun datasets. Following previous DG methods (Li et al., 2019; Zhou et al., 2021c; Li et al., 2018b;a), we evaluate methods under the leave-one-domain-out protocol, where one domain is used for testing while others for training.

**Implementation Details.** For PACS and Office-Home, we take ResNet18 (He et al., 2016) pretrained on ImageNet (Deng et al., 2009) as backbone, following Zhou et al. (2021c); Carlucci et al. (2019). We also take the ResNet-50 pretrained on ImageNet as the backbone for PACS, following Huang et al. (2020). For VLCS, we take AlexNet (Krizhevsky et al., 2012) pretrained on ImageNet (Deng et al., 2009) as our backbone, which is the same as (Matsuura & Harada, 2020; Dou et al., 2019). For Digits-DG, we adopt the model architecture used in previous works (Zhou et al., 2020b; 2021c). We cluster three clusters within each class in training datasets. All reported results are averaged among six runs. More implementation details can be found in the **Appendix** A.5.

## 4.2 ABLATION STUDY

**Effectiveness of *CICF*.** Our proposed *CICF* enables models to have superior generalization ability by effectively causal feature learning. We compare our *CICF* with the popular meta-learning strategy MAML, which aims to explore the commonly optimal optimization direction for all tasks (*i.e.*, different domains), on two baselines. 1) ERM: training models only on source domains with simple data augmentations including flip and translation. 2) ERM*: training models on source domains with AutoAugment (Cubuk et al., 2018). The results on PACS are shown in Table 1, with baseline ERM, ERM+*CICF* outperforms ERM by 2.4 % in accuracy *without known domain labels*, while MAML only achieves the improvement of 0.1% *with known domain labels*. Moreover, with another baseline, ERM*+*CICF* outperforms ERM* by 3.2% and 2.0% on PACS and Digits-DG respectively. However, MAML is not robust for different baselines and does not work for ERM*.

| Method | A | C | P | S | Avg. |
|---|---|---|---|---|---|
| MatchDG | 85.6 | 82.1 | 97.9 | 78.8 | 86.1 |
| RSC | 87.9 | **82.2** | 97.9 | 83.4 | 87.8 |
| FACT | 89.6 | 81.8 | 96.8 | 84.5 | 88.2 |
| Fish | - | - | - | - | 85.5 |
| ERM* | 88.0 | 78.8 | **98.2** | 81.7 | 86.7 |
| ERM*+*CICF* | **89.7** | **82.2** | 97.9 | **86.2** | **89.0** |

Table 2: Classification accuracy (%) of different DG methods on PACS with ResNet-50.

**Different ways to estimate global-scope intervening effects.** To estimate the global-scope intervening effects efficiently and accurately, we propose a *clustering-then-sampling* strategy. An alternative is the naïve random mini-batch sampling. As shown in Fig. 3(a), the classification accuracy increases along with the increased sampling batch-size. This is because increasing batch-size will result in a better estimation of the global-scope intervention with lower SE. However, the memory overhead increases drastically simultaneously. In contrast, our *clustering-then-sampling* outperforms the naïve random sampling by 0.7% with lower memory utilization (the batch-size is fixed as

256 for PACS in our case). We also conduct experiments on the influence of the number of clusters $K^{\dagger}$ for all datasets. The results in Fig. 3(b) show that more clusters result in better performance, and the performance is saturated when $K^{\dagger} = 21$ for PACS (*i.e.*, three clusters for each class). Because more clusters lead to lower intra-cluster variance $\sigma_k$ of the $k^{th}$ cluster and more accurate estimation of the global-scope intervention with lower SE, which is derived in the **Appendix** A.1.1. And three clusters for each class are enough for global-scope intervention modeling.

**Different choices of $Z$.** We explore the effects of different choices of $Z$ from shallow layer features to deep layer features. The experiments and analysis are shown in Appendix A.4, which reveals that shallow features are better.

### 4.3 COMPARISON WITH STATE-OF-THE-ARTS

**PACS.** As shown in Table 1, our proposed *CICF* on top of ERM$^*$ achieves the best performance on PACS, outperforming the SOTA works including MixStyle (Zhou et al., 2021c), RSC (Huang et al., 2020), MatchDG (Mahajan et al., 2021a) and FACT Xu et al. (2021) Moreover, ERM$^*$+*CICF* is clearly better than previous gradient-based and meta-learning based methods, *e.g.*, CrossGrad (Shankar et al., 2018), MLDG (Li et al., 2018a), MetaReg (Balaji et al., 2018), and MAML (Finn et al., 2017), thanks to the more accurate global intervening effects modeling in *CICF*. On the most challenging domain *sketch*, our ERM$^*$+*CICF* outperforms all previous methods by a large margin ($\geq$ **2.3%**), which demonstrates that *CICF* can learn the causal visual features by removing the influence from confounders efficiently. The experiments on PACS with ResNet-50 are shown in Table 2, where our *CICF* is clearly superior to the SOTA methods, *e.g.,* RSC (Huang et al., 2020), MatchDG (Mahajan et al., 2021a), FACT (Xu et al., 2021) and recent Fish (Shi et al., 2021).

**Digits-DG.** As shown in Table. 1, our ERM$^*$+*CICF* achieves the best performance, outperforming the MAML (Finn et al., 2017) by 1.4% and the recent MixStyle (Zhou et al., 2021c) by 5.2%. For the most challenging domains (*i.e.*, MNIST-M and SVHN), ERM$^*$+*CICF* improves the accuracy of ERM$^*$ by 3.8% and 2.5% respectively.

**Office-Home.** As shown in Table 3 of Appendix, our ERM$^*$+*CICF* achieves the best performance of 66.2%, exceeding the previous state-of-the-art method MixStyle (Zhou et al., 2021c) by 0.7%. Furthermore, ERM$^*$+*CICF* improves ERM$^*$ on the challenging *Clipart* by a margin of 3.2%.

**VLCS.** The results on VLCS are shown in Table 4 of Appendix, where our ERM$^*$+*CICF* achieves the state-of-the-art result, outperforming the second best MASF (Dou et al., 2019) by 0.6%, which is based on meta-learning. Moreover, *CICF* brings the improvement of 1.7% for ERM$^*$.

### 4.4 FEATURE VISUALIZATION

To validate that *CICF* actually learns the causal visual features, we visualize and compare the Grad-CAM (Selvaraju et al., 2017) of ERM and ERM+*CICF* in Fig. 4. Intervened by the confounders (*e.g.*, background), ERM easily focuses on the object-irrelevant regions (*i.e.*, non-causal features), impeding the model's generalization ability. In contrast, thanks to the guidance of *CICF*, ERM+*CICF* is prone to focus more on the foreground object regions (*i.e.*, causal features). Further, we visualize the learned features by t-SNE (Saito et al., 2019) on Digits-DG in the **Appendix** A.6.

### 5 CONCLUSION

In this paper, we propose a novel method dubbed *Confounder Identification-free Causal Visual Feature Learning* (*CICF*) for learning causal visual features without explicit identification and exploitation of confounders. Particularly, motivated and based on the front-door criterion, we model the interventions among samples and approximate the global-scope intervening effects for causal visual feature learning. Extensive experimental results on domain generalization validate that our *CICF* can help a model to achieve superior generalization capability by learning causal features, without the need of identifying confounders. Our method is generic which should be applicable to other fields such as NLP. We leave this as future work.

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

# A APPENDIX

## A.1 MORE DETAILS ON *CICF*

### A.1.1 STATISTICAL ANALYSIS ON SAMPLING ALGORITHMS

In this section, we provide the statistical analysis for the superiority of our proposed sampling strategy, *i.e.*, *clustering-then-sampling*, by comparing it to the *random mini-batch sampling* strategy. As a result, we theoretically derive that using our *clustering-then-sampling* has a significant smaller standard error (SE) in estimating the global-scope gradient, compared with using the *random mini-batch sampling* strategy.

Given a training dataset $D = (x_i, y_i)_{i=1}^{N}$, the global-scope gradient can be computed as:

$$g_{\dagger} = \sum_{i=1}^{N} g_i P(x_i), where \; P(x_i) = \frac{1}{N} \tag{12}$$

However, it is intractable to compute $g_{\dagger}$ with Eq. 12 directly, which requires traversing over all training data.

Our *clustering-then-sampling* aims to simulate the global-scope gradient with partial data from the training dataset. Another alternative naïve method is *random mini-batch sampling.*

***Random mini-batch sampling.*** With this strategy, the gradient $g_{random}$ of each iteration is computed over a mini-batch data $\{x_j^r, y_j^r\}_{j=1}^{M}$, where the mini-batch data with size $M$ is randomly sampled from the training data:

$$g_{random} = \sum_{j=1}^{M} g_j P(x_j^r), where \; P(x_j^r) = \frac{1}{M} \tag{13}$$

According to the sampling theory (Ghosh, 2002), the expectation of $g_{random}$ can be represented as:

$$E(g_{random}) = E(\frac{1}{M} \sum_{j=1}^{M} g_j) = \frac{1}{M} \sum_{j=1}^{M} E(g_j) = \mu, \tag{14}$$

where $\mu$ denotes the expectation of gradients of all samples in the training data. To measure this strategy's estimation accuracy of approximating the global-scope gradient $g_{random}$ in Eq. 12, we can derive its SE as:

$$SE_{random} = \frac{\sigma^2}{M}(1 - \frac{M-1}{N-1}), \tag{15}$$

where $\sigma^2$ is the variance of the gradients of all training samples. Considering $M \ll N$, Eq. 15 can be further approximated as:

$$SE_{random} \approx \frac{\sigma^2}{M}. \tag{16}$$

***clustering-then-sampling.*** In contrast, in our clustering-then-sampling, we first cluster the training samples of each class in the dataset into $K$ cluster with $K$-means algorithm (Pelleg et al., 2000) and totally obtain $K^{\dagger}$ clusters for the whole training data. We denote the mean and variance of the gradients of the $k^{th}$ cluster $K_k$ as $\mu_k$ and $\sigma_k^2$ respectively. Since the clustering, the variance of the sample distribution in each cluster is significantly smaller than the variance of the sample distribution in whole training data. Therefore, the gradient variance of the $k^{th}$ cluster $\sigma_k^2$ is smaller than $\sigma^2$. Then we sample $N_k$ samples from the $k^{th}$ cluster to form a mini-batch with the size of $M$ (*i.e.*, $M = \sum_{k=1}^{K^{\dagger}} N_k$). Here, $N_k$ is proportional to the ratio $P(K_k)$ (*i.e.*, $N_k = MP(K_k)$) and $P(K_k)$ is ratio of the size of the $k^{th}$ cluster $N_k^K$ to the size of entire training data $N$ (*i.e.*, $P(K_k) = \frac{N_k^K}{N}$). Then the global-scope gradient can be computed with:

$$g_{ours} = \frac{1}{M} \sum_{k=1}^{K^{\dagger}} \sum_{j=1}^{N_k} g_{\tilde{x}_{j,k} \in K_k} \tag{17}$$

---

**Algorithm 1** Confounder Identification-free Causal Visual Feature Learning

---

1: **Input:** Training dataset $\{(x_i, y_i)\}_{i=1}^N$.
2: **Init:** learning rate: $\alpha$, $\beta$; model $f$ with parameter $\theta$.
3: Obtain $K^\dagger$ clusters from training data by clustering the samples of each class into $K$ clusters.
4: **while** not converge **do**
5:      Sample $M$ samples from $K^\dagger$ clusters as a batch.
6:      Estimate global intervention with $g_\dagger$.                  ▷ Eq. 7
7:      Update $f$ with $g_\dagger$ as: $\theta_\dagger = \theta - \alpha g_\dagger$.
8:      Compute the loss $\mathcal{L}_{CICF}$.                         ▷ Eq. 10
9:      Update $\theta \leftarrow \theta - \beta \nabla_\theta \mathcal{L}_{CICF}$.
10: **end while**

---

Based on stratified sampling theory (Ghosh, 2002), the expectation of $g_{ours}$ can be computed as:

$$E(g_{ours}) = \sum_{k=1}^{K^\dagger} \frac{N_k}{M} \mu_k = \sum_{k=1}^{K^\dagger} \frac{MP(K_k)}{M} \mu_k = \mu \tag{18}$$

The standard error (SE) of our *clustering-then-sampling* can be derived as :

$$SE_{ours} = \sum_{k=1}^{K^\dagger} P(K_k)^2 \frac{1}{N_k} (1 - \frac{N_k - 1}{N_k^K - 1}) \sigma_k^2, \tag{19}$$

where $N_k^K$ denotes the number of all samples in the $k^{th}$ cluster. Considering $N_k \ll N_k^K$,

$$SE_{ours} \approx \sum_{k=1}^{K^\dagger} P(K_k)^2 \frac{1}{N_k} \sigma_k^2 = \sum_{k=1}^{K^\dagger} \frac{P(K_k)}{M} \sigma_k^2 \tag{20}$$

In clustering, the intra-cluster gradient variance $\sigma_k^2$ reduces with the increasing the number of clusters. Thus, we can get $\sigma_k^2 < \sigma^2$ when $1 \le k \le K^\dagger$. We represent the maximum value of $\sigma_k^2$ as $(\sigma_k^2)_{max} = max\{\sigma_k^2 | 1 \le k \le K^\dagger\}$. The Eq. 20 can be rewritten as:

$$SE_{ours} = \sum_{k=1}^{K^\dagger} \frac{P(K_k)}{M} \sigma_k^2 \le (\sigma_k^2)_{max} \sum_{k=1}^{K^\dagger} \frac{P(K_k)}{M}$$

$$= \frac{1}{M} (\sigma_k^2)_{max} < \frac{1}{M} \sigma^2 = SE_{random} \tag{21}$$

Based on the Eq. 21, we can draw a conclusion that our *clustering-then-sampling* is better than *random mini-batch sampling* for simulating the global-scope gradient accurately. Furthermore, from the Eq. 21, we can let $(\sigma_k^2)_{max} \ll \sigma^2$ by increasing the number of clusters, and obtain the $SE_{ours} \ll SE_{random}$.

### A.1.2 EXPERIMENTAL EVIDENCE FOR THE SIGNIFICANT DIFFERENCE BETWEEN TWO SAMPLING STRATEGIES.

To further demonstrate the difference between our *clustering-then-sampling* and *random mini-batch sampling*, we introduce a metric $E = \sum_{k=1}^{K^\dagger} (|N_k - R_k|)$ to measure the difference degree between two sampling strategies, where $N_k$ and $R_k$ denote the number of sampled instances from the $k^{th}$ cluster using *clustering-then-sampling* and *random mini-batch sampling*, respectively. We conduct two experiments on PACS with batch-size $M = 256$ as follows. 1) We cluster three clusters for each of the seven classes in PACS and totally obtain 21 clusters. The average E is 55, *i.e.*, the difference ratio $E/M$ between *clustering-then-sampling* and *random mini-batch sampling* is $55/256 = 21.5\%$. 2) We consider the class prior for *random mini-batch sampling*, and adopt the *random mini-batch sampling* weighted by the number of each class. We have $E$ as 46 and the difference ratio $E/M = 46/256 = 18.0\%$. Based on the above experiments, we can find that our *clustering-then-sampling* is significantly different from *random mini-batch sampling*.

### A.1.3 More Details on the Setting of Sampling Algorithm

Our *CICF* aims to simulate the global intervening effects from the perspective of optimization with:

$$g_\dagger = \frac{1}{M} \sum_{k=1}^{K^\dagger} \sum_{j=1}^{N_k} g_{\tilde{x}_{j,k} \in K_k}, \tag{22}$$

where a mini-batch sampling procedure is applied to $\tilde{x}_{j,k} \in K_k$ (*i.e.*, clustering-then-sampling). After obtaining the updated $\theta_\dagger$ with $\theta_\dagger = \theta - \alpha * g_\dagger$, we can obtain the loss function $\mathcal{L}_{CICF}$ and get the reliable optimization direction against the intervening effects of confounders:

$$\mathcal{L}_{CICF} = \sum_{x \in X} \mathcal{L}_{ce}(f_{\theta_\dagger}(h(x)), y), \tag{23}$$

where another mini-batch sampling procedure is applied to the variable $x$.

As above, there are two mini-batch sampling in our *CICF* adopted for computing global-scope gradient and loss function $L_{CICF}$ respectively. They are denoted by $\{\tilde{x}, \tilde{y}\}_{i=1}^M$ and $\{x, y\}_{i=1}^{M^l}$, respectively. To clarify the composition of the samples in each mini-batch, we first define some notations, respectively as follows:

- $N$: The number of all samples in the training dataset
- $N_k^K$: The number of all samples in the $k^{th}$ cluster.
- $N_k$: The number of sampled samples in the $k^{th}$ cluster to form a mini-batch.
- $M$: The number of all samples in a mini-batch for computing global-scope gradient.
- $M^l$: The number of all samples in a mini-batch for computing $L_{CICF}$.
- $P(K_k)$: The ratio of all samples $N_k^K$ in the $k^{th}$ cluster to all training data $N$, which is represented as $P(K_k) = \frac{N_k^K}{N}$

For computing global-scope gradient, we respectively sample $N_k$ samples from the $k^{th}$ cluster to form a mini-batch $\{\tilde{x}, \tilde{y}\}_{i=1}^M$. $N_k$ can be set with two strategies, respectively $N_k = MP(K_k)$ and $N_k = \frac{M}{K^\dagger}$, responding to two scheme for computing $L_{CICF}$.

When we directly consider the unbalance question between different clusters in computing $L_{CICF}$, we can sample $\frac{M^l}{K^\dagger}$ samples from each cluster to form a mini-batch $\{x, y\}_{i=1}^{M^l}$, and corresponding $N_k$ is $N_k = \frac{M}{K^\dagger}$. On the contrary, when we randomly sample $M^l$ samples from all training data to compute $L_{CICF}$, the corresponding $N_k$ is $N_k = MP(K_k)$, which needs to model the intervention caused by the unbalance question.

### A.2 More Details on Back-door/Front-door Criterion

Back-door and front-door criteria have been proposed in Pearl et al. (2016); Pearl (2009a) to reveal the causality between two variables $X$ and $Y$. We further clarify their basics mathematically in this section.

**Back-door criterion.** Fig. 5 shows the back-door criterion, which targets for removing the intervention effects with *do* operation. *Do* operation denotes a surgery to cut off the connection from $C$ to $X$. From Fig. 5(a), the $P(Y|X)$ is associated with two paths, respectively as $X \leftarrow C \rightarrow Y$ and $X \rightarrow Y$. Here $X \leftarrow C \rightarrow Y$ is a spurious path, which intervenes the estimation of causality between $X$ and $Y$, denoted as $P(Y|do(X))$ (*i.e.*, the Fig. 5(b)). Following the Pearl et al. (2016), we can denote the conditional probability between $X$ and $Y$ in Fig. 5(b) as $P_m(Y|X)$. Since $C$ has no parent-level variable, the $P_m(C) = P_m(C|X)$ in the Fig. 5(b) is equivalent to the $P(C)$ in the Fig. 5(a). Furthermore, we can get the $P_m(Y|X, C)$ in the Fig. 5(b) is equivalent to $P(Y|X, C)$ in the Fig. 5(a) since the same graph architecture. Based on the above definition, the back-door

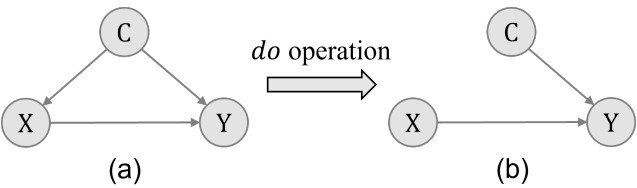

Figure 5: Back-door criterion

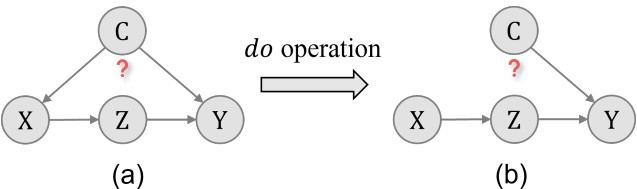

Figure 6: Front-door criterion

criterion can be derived as:

$$P(Y|do(X)) = P_m(Y|X)$$
$$= \sum_c P_m(Y|X, C = c)P_m(C = c|X)$$
$$= \sum_c P_m(Y|X, C = c)P_m(C = c)$$
$$= \sum_c P(Y|X, C = c)P(C = c) \tag{24}$$

Then we can get the formulation of the back-door criterion as:

$$P(Y|do(X)) = \sum_c P(Y|X, C = c)P(C = c). \tag{25}$$

**Front-door criterion.** From Eq. 24, we can draw a conclusion that estimating the causality between $X$ and $Y$ (*i.e.*, $P(Y|do(X))$) requires traversing the distribution of confounders $C$. However, confounders $C$ are in general diverse and not identifiable. To solve the above question, as shown in Fig. 6, the front-door criterion introduces an intermediate variable $Z$ and transfers the requirement of modeling the intervening effects of confounders $C$ on $X \to Y$ to modeling the intervening effects of $X$ on $Z \to Y$ (Pearl et al., 2016). Specifically, front-door criterion decompose the $P(Y|do(X))$ into two components, *i.e.*, $P(Y|do(Z))$ and $P(Z|do(X))$, which can be represented as:

$$P(Y|do(X)) = \sum_z P(Z = z|do(X))P(Y|do(Z = z)) \tag{26}$$

As shown in Fig. 6(a), the variables $X$ and $Z$ do not have common causes, which reveals the path $X \to Z$ is not intervened by other variables. Therefore, the causality between $X$ and $Z$ is equivalent to its correlation as:

$$P(Z = z|do(X)) = P(Z = z|X) \tag{27}$$

From Fig. 6(a), the path $Z \to Y$ is intervened by two variables, respectively as $C$, $X$, since the existing spurious path $Z \leftarrow X \leftarrow C \to Y$. Then we can simply block this spurious path by cutting off the path $X \to Z$ with back-door criterion (Pearl et al., 2016), which gets rid of identifying the confounders $C$.

$$P(Y|do(Z = z)) = \sum_{\tilde{x} \in X} P(Y|Z = z, \tilde{x})P(\tilde{x}) \tag{28}$$

Based on Eq. 27 and 28, we can derive Eq. 26 as:

$$P(Y|do(X)) = \sum_z P(Z = z|X) \sum_{\tilde{x} \in X} P(Y|Z = z, \tilde{x})P(\tilde{x}) \tag{29}$$

Then we can obtain the formulation of the front-door criterion as Eq. 29.

A.3 USING $h(X)$ AS $Z$ SATISFIES FRONT-DOOR CRITERION

In this section, we give the proof that using $h(x)$ as $Z$ satisfies the front-door criterion. As depicted in Pearl et al. (2016), the definition of the front-door criterion is as following:

**Definition:** *if $Z$ satisfies the front-door criterion relative to an ordered pair of variables $(X, Y)$, it must obey the following principles: 1) $Z$ intercepts all directed paths from $X$ to $Y$. 2) There is no unblocked back-door path from $X$ to $Z$. 3) All back-door paths from $Z$ to $Y$ are blocked by $X$.*

We prove that ours satisfy the above principles as:

- As shown in Fig. 2, the direct path from $X$ to $Y$ is built from the model $\varphi$, that is composed of two sub-models, $h$ and $f$. We can represent this path as $Z = g(X), Y = f(Z)$. We can observe that $Z$ intercepts all directed paths from $X$ to $Y$, which satisfies the ***Principle 1)***.

- The second principle requires there is no unblocked back-door path from $X$ to $Z$, (*i.e.,* $P(Z|do(X)) = P(Z|X)$), which is a vital factor in ensuring that the causal effects can be estimated. We interpret that ours satisfy this principle from two perspectives. 1) As shown in Fig. 2, the effect of confounders C to Z is transmitted by X through a sub-model $f$ as $C \to X \xrightarrow{f} Z$ instead of $X \leftarrow C \to Z$, which indicates that there is no back-door from X to Z and thus no common confounders for X and Z. 2) We prove this from contradiction. From the theory of the back-door criterion, if there are common confounders C for X and Z, then $P(Z|do(X))! = P(Z|X)$. In fact, when $Z = h(X)$ is obtained from a deterministic mapping of X at the inference stage, we always have $P(Z|X, C) = P(Z|X)$ for any confounder C. Then we can derive that $P(Z|do(X)) = \sum_{C=c} P(Z|X, c)P(c) = \sum_{C=c} P(Z|X)P(c) = P(Z|X)$. This means there are no common confounders for X and Z (*i.e.,* no unblocked back-door path for X and Z), which satisfies the ***Principal 2)***.

- There are two paths from $Z$ to $Y$, *i.e.,* $Z \to Y$ and $Z \leftarrow X \leftarrow C \to Y$. The second path is called the back-door path between $Z$ and $Y$. When we condition on $X$, the back-door path will be blocked. Therefore, all back-door paths from $Z$ to $Y$ are blocked by $X$, which satisfies the ***Principal 3)***.

Consequently, exploiting $Z = h(X)$ is consistent with the front-door criterion.

Another essential factor for estimating the causal effects by the front-door criterion is $P(X, Z) > 0$. That means if $P(X, Z)=0$, the $P(Y|do(X))$ in Eq. 2 is equivalent 0 and cannot be estimated. In our **CICF**, $Z$ is deterministic related to $X$ with $Z = f(X)$. Based on the probability theory $P(X, Z) = P(X)P(Z|X)$. Since $Z = f(X)$, the $P(Z|X) = P(f(X)|X) = 1$ is always holds when $f$ is fixed at inference stage. Therefore, $P(X, Z) = P(X)P(Z|X) = P(X) > 0$ holds in our method.

A.4 DIFFERENT CHOICES OF $Z$

As shown in Fig. 2 (a), the model $\varphi$ is separated into successive $h$ and $f$, and $Z = h(X)$ is the intermediate output of $\varphi$. To explore the effects of different choices of $Z$, *i.e.*, and different separations of $\varphi$, we conduct ablation experiments on PACS. Fig. 7 shows the different choices of $Z$ based on ResNet and the corresponding results are shown in Fig. 3(c). It is observed that the shallower $Z$ is obtained, the better accuracy the results achieved. We reckon the reason is that selecting $Z$ from shallower layers will result in a larger model for $f$, which will have more capability to learn the conditional probability $P(Y|do(Z))$. In this paper, we set $Z = Z_0$ to make $f$ have more parameters to learn causal features.

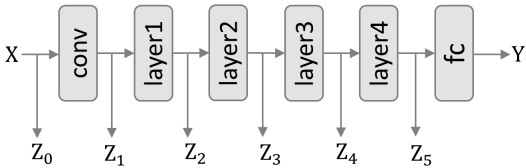

Figure 7: Different choices for the intermediate output $Z$.

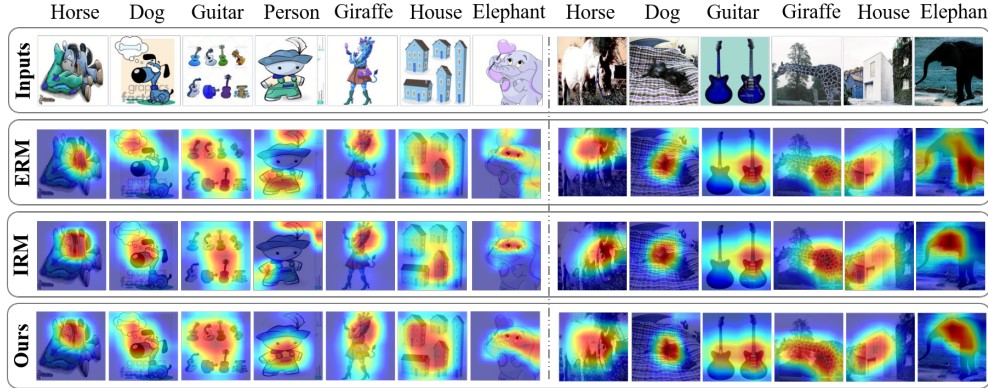

Figure 8: Visualization of the Grad-Cams w.r.t. classification task for domain generalization. The first row of left/right panels shows the images from cartoon/photo domains in PACS. The second, third and fourth rows show the Grad-Cams of ERM, IRM (Arjovsky et al., 2019) and our *CICF*.

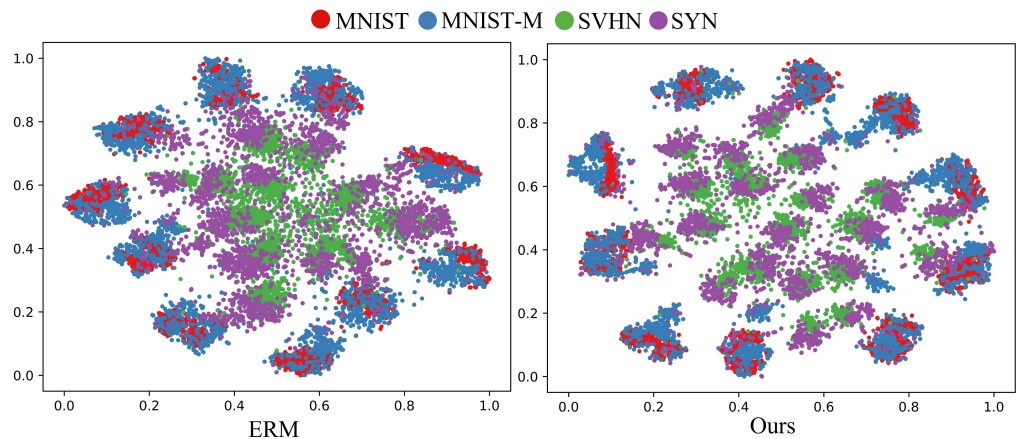

Figure 9: t-SNE visualization of features learned by ERM (left) and our *CICF* (right).

## A.5  MORE IMPLEMENTATION DETAILS

For PACS and Office-Home, we take ResNet18 (He et al., 2016) pretrained on ImageNet (Deng et al., 2009) as backbone, following Zhou et al. (2021c); Carlucci et al. (2019); Li et al. (2018a). For VLCS, we take AlexNet (Krizhevsky et al., 2012) pretrained on ImageNet (Deng et al., 2009) as our backbone, which is the same as Dou et al. (2019); Li et al. (2019); Matsuura & Harada (2020). For Digits-DG, we adopt the model architecture used in previous works (Zhou et al., 2020b; Carlucci et al., 2019; Zhou et al., 2021c), which is composed of four convolution layers with inserted ReLU and max-pooling layers. We set the mini-batch size $M$ for computing the global-scope gradient as 256, and the mini-batch size $M^l$ for computing $L_{CICF}$ as 84. We set the batch size to 84 and train the model using SGD optimizer for 60 epochs. For PACS, the learning rate $\alpha$ and $\beta$ are set to 0.05 and 0.01 respectively. For VLCS, the learning rate $\alpha$ and $\beta$ are 0.005 and 0.001 respectively. As for Office-Home, we set $\alpha$, $\beta$ as 0.001 and 0.001 respectively. For Digits-DG, we set $\alpha$, $\beta$ as 0.5 and 0.1 respectively. All reported results are averaged among six runs with different seeds.

The GPU memory cost of our *CICF* is 7.151 GiB and the clustering for training data takes 164 seconds for PACS, which introduces only $4.8\%$ extra time to the whole training process (*i.e.*, 3432 seconds) while bringing the significant gain of 2.4%.

Table 3: Classification accuracy (%) of different DG methods on Office-Home with ResNet-18 as the backbone.

| Method | Office-Home | | | | |
|---|---|---|---|---|---|
| | Art | Clipart | Product | Realworld | Avg. |
| JiGen | 53.0 | 47.5 | 71.5 | 72.8 | 61.2 |
| MMD-AAE | 56.5 | 47.3 | 72.1 | 74.8 | 62.7 |
| MLDG | 57.8 | 50.3 | 70.6 | 73.0 | 63.0 |
| CrossGrad | 58.4 | 49.4 | 73.9 | 75.8 | 64.4 |
| CCSA | **59.9** | 49.9 | 74.1 | 75.7 | 64.9 |
| MixStyle | 58.7 | 53.4 | 74.2 | **75.9** | 65.5 |
| ERM | 58.1 | 48.7 | 74.0 | 75.6 | 64.2 |
| ERM+MAML | 56.8 | 52.5 | 74.0 | 74.7 | 64.5 |
| ERM+*CICF* | 57.1 | 52.0 | 74.1 | 75.6 | 64.7 |
| ERM* | 59.6 | 53.0 | **74.3** | 75.4 | 65.6 |
| ERM*+MAML | 56.2 | 56.1 | 72.6 | 73.2 | 64.5 |
| ERM*+*CICF* | 59.3 | **56.2** | 74.2 | 75.1 | **66.2** |

Table 4: Classification accuracy (%) of different DG methods on VLCS with AlexNet as the backbone.

| Methods | VLCS | | | | |
|---|---|---|---|---|---|
| | Caltech | Labelme | Pascal | Sun | Avg. |
| MLDG | 97.9 | 59.5 | 66.4 | 64.8 | 72.2 |
| Epi-FCR | 94.1 | 64.3 | 67.1 | 65.9 | 72.9 |
| JiGen | 96.93 | 60.9 | 70.6 | 64.3 | 73.2 |
| MMLD | 96.6 | 58.7 | **72.1** | 66.8 | 73.5 |
| MASF | 94.8 | **64.9** | 69.1 | 67.6 | 74.1 |
| ERM | 96.3 | 59.7 | 70.6 | 64.5 | 72.8 |
| ERM+MAML | 97.8 | 58.0 | 67.1 | 64.1 | 71.8 |
| ERM+*CICF* | 97.8 | 60.1 | 69.7 | 67.3 | 73.7 |
| ERM* | 96.4 | 60.7 | 68.6 | 66.2 | 73.0 |
| ERM*+MAML | 98.1 | 58.2 | 69.6 | 64.5 | 72.6 |
| ERM*+*CICF* | **98.1** | 62.4 | 69.3 | **69.1** | **74.7** |

## A.6 FEATURE VISUALIZATION

We visualize more Grad-CAM (Selvaraju et al., 2017) of ERM and ERM+*CICF* in Fig. 8. We can observe that our *CICF* focus more on foreground regions (*i.e.*, the casual features), while ERM easily focuses on the misleading regions (*e.g.*, the bone in the dog of the cartoon, background) when capturing causal features. As shown in Fig. 9, we visualize the learned feature on Digits-DG **by t-SNE** (Saito et al., 2019). We find that the distribution of features extracted from ERM+*CICF* is more compact across samples with the same category, compared to ERM. This validates the effectiveness of our algorithm for causal feature learning. We also visualize the t-SNE visualizations with different random seeds in the Fig. 10 and Fig. 11.

## A.7 MORE COMPARISONS WITH THE DOMAINBED BENCHMARKS.

To further demonstrate the effectiveness of our *CICF*, we implement our *CICF* on the baseline ERM of DomainBed (Gulrajani & Lopez-Paz, 2020) and compare it with the benchmarks used in DomainBed (Gulrajani & Lopez-Paz, 2020), including IRM (Arjovsky et al., 2019), DRO (Sagawa et al., 2019), Mixup (Xu et al., 2020), MLDG (Li et al., 2018a), CORAL (Sun et al., 2016), MMD (Li et al., 2018b), DANN (Ganin et al., 2016), and C-DANN (Li et al., 2018c). We utilize the commonly-used datasets Office-Home (Venkateswara et al., 2017), VLCS (Torralba & Efros, 2011), and the synthesized ColorMNIST (Arjovsky et al., 2019). As shown in the Table 5, Table 6, Table 7, our *CICF* improves the ERM by an average gain of 2% on Office-Home, 1.7% on VLCS and 4.1% on ColorMNIST. Our *CICF* achieves the best performances on these three datasets. It is noteworthy

that even in the most difficult domain in ColorMNIST, our *CICF* can achieve an accuracy of 21.4%, which exceeds the second method MMD (Li et al., 2018b) by 10.9%.

Table 5: Classification accuracy (%) of different DG methods on Office-Home with ResNet-50 as the backbone.

| Office-Home: Model selection method: training domain validation set | | | | | |
|---|---|---|---|---|---|
| Algorithm | A | C | P | R | Avg. |
| ERM | 62.7 ± 1.1 | 53.4 ± 0.6 | 76.5 ± 0.4 | 77.3 ± 0.3 | 67.5 ± 0.5 |
| IRM | 61.8 ± 1.0 | 52.3 ± 1.0 | 75.2 ± 0.8 | 77.2 ± 1.1 | 66.6 ± 1.0 |
| DRO | 61.6 ± 0.7 | 52.9 ± 0.2 | 75.5 ± 0.5 | 77.7 ± 0.2 | 66.9 ± 0.3 |
| Mixup | **64.7 ± 0.7** | 54.7 ± 0.6 | 77.3 ± 0.3 | **79.2 ± 0.3** | 69.0 ± 0.1 |
| MLDG | 63.7 ± 0.3 | 54.5 ± 0.6 | 75.9 ± 0.4 | 78.6 ± 0.1 | 68.2 ± 0.1 |
| CORAL | 64.4 ± 0.3 | 55.3 ± 0.5 | 76.7 ± 0.5 | 77.9 ± 0.5 | 68.6 ± 0.4 |
| MMD | 63.0 ± 0.1 | 53.7 ± 0.9 | 76.1 ± 0.3 | 78.1 ± 0.5 | 67.7 ± 0.1 |
| DANN | 59.3 ± 1.1 | 51.7 ± 0.2 | 74.1 ± 0.8 | 76.6 ± 0.6 | 65.4 ± 0.6 |
| C-DANN | 61.0 ± 1.4 | 51.1 ± 0.7 | 74.1 ± 0.3 | 76.0 ± 0.7 | 65.6 ± 0.5 |
| ERM + CICF (Ours) | 63.1 ± 0.2 | **59.4 ± 0.2** | **77.4 ± 0.0** | 78.1 ± 0.4 | **69.5 ± 0.2** |

Table 6: Classification accuracy (%) of different DG methods on VLCS with ResNet-50 as the backbone

| VLCS: Model selection method: training domain validation set | | | | | |
|---|---|---|---|---|---|
| Algorithm | C | L | S | V | Avg. |
| ERM | 97.6 ± 1.0 | 63.3 ± 0.9 | 72.2 ± 0.5 | 76.4 ± 1.5 | 77.4 ± 0.3 |
| IRM | 97.6 ± 0.3 | 65.0 ± 0.9 | 72.9 ± 0.5 | 76.9 ± 1.3 | 78.1 ± 0.0 |
| DRO | 97.7 ± 0.4 | 62.5 ± 1.1 | 70.1 ± 0.7 | 78.4 ± 0.9 | 77.2 ± 0.6 |
| Mixup | 97.9 ± 0.3 | 64.5 ± 0.6 | 71.5 ± 0.9 | 76.9 ± 1.3 | 77.7 ± 0.4 |
| MLDG | 98.1 ± 0.3 | 63.0 ± 0.9 | 73.5 ± 0.6 | 73.7 ± 0.3 | 77.1 ± 0.4 |
| CORAL | 98.8 ± 0.1 | 64.6 ± 0.8 | 71.7 ± 1.4 | 75.8 ± 0.4 | 77.7 ± 0.5 |
| MMD | 97.1 ± 0.4 | 63.4 ± 0.7 | 71.4 ± 0.8 | 74.9 ± 2.5 | 76.7 ± 0.9 |
| DANN | 98.5 ± 0.2 | 64.9 ± 1.1 | 73.1 ± 0.7 | 78.3 ± 0.3 | 78.7 ± 0.3 |
| C-DANN | 97.5 ± 0.1 | **65.2 ± 0.4** | 73.4 ± 1.1 | 76.9 ± 0.2 | 78.2 ± 0.4 |
| ERM + CICF (Ours) | **99.1 ± 0.4** | 64.0 ± 0.1 | **74.2 ± 0.5** | **79.2 ± 0.4** | **79.1 ± 0.3** |

Table 7: Classification accuracy (%) of different DG methods on ColorMNIST with MNIST backbone used in DomainBed

| ColorMNIST: Model selection method: training domain validation set | | | | |
|---|---|---|---|---|
| Algorithm | 0.1 | 0.2 | 0.9 | Avg. |
| ERM | 72.7 ± 0.2 | 73.2 ± 0.3 | 10.0 ± 0.0 | 52.0 ± 0.1 |
| IRM | 72.0 ± 0.3 | 73.2 ± 0.0 | 10.1 ± 0.2 | 51.8 ± 0.1 |
| DRO | 72.7 ± 0.3 | 73.1 ± 0.3 | 10.0 ± 0.0 | 52.0 ± 0.1 |
| Mixup | 72.4 ± 0.2 | 73.3 ± 0.3 | 10.0 ± 0.1 | 51.9 ± 0.1 |
| MLDG | 71.4 ± 0.4 | 73.3 ± 0.0 | 10.0 ± 0.0 | 51.6 ± 0.1 |
| CORAL | 71.8 ± 0.4 | 73.3 ± 0.2 | 10.1 ± 0.1 | 51.7 ± 0.1 |
| MMD | 72.1 ± 0.2 | 72.8 ± 0.2 | 10.5 ± 0.2 | 51.8 ± 0.1 |
| ADA | 72.0 ± 0.3 | 72.4 ± 0.5 | 10.0 ± 0.2 | 51.5 ± 0.3 |
| CondADA | 72.2 ± 0.3 | 73.2 ± 0.2 | 10.4 ± 0.3 | 51.9 ± 0.1 |
| ERM + CICF (Ours) | **72.9 ± 0.3** | **74.1 ± 0.2** | **21.4 ± 0.4** | **56.1 ± 0.2** |

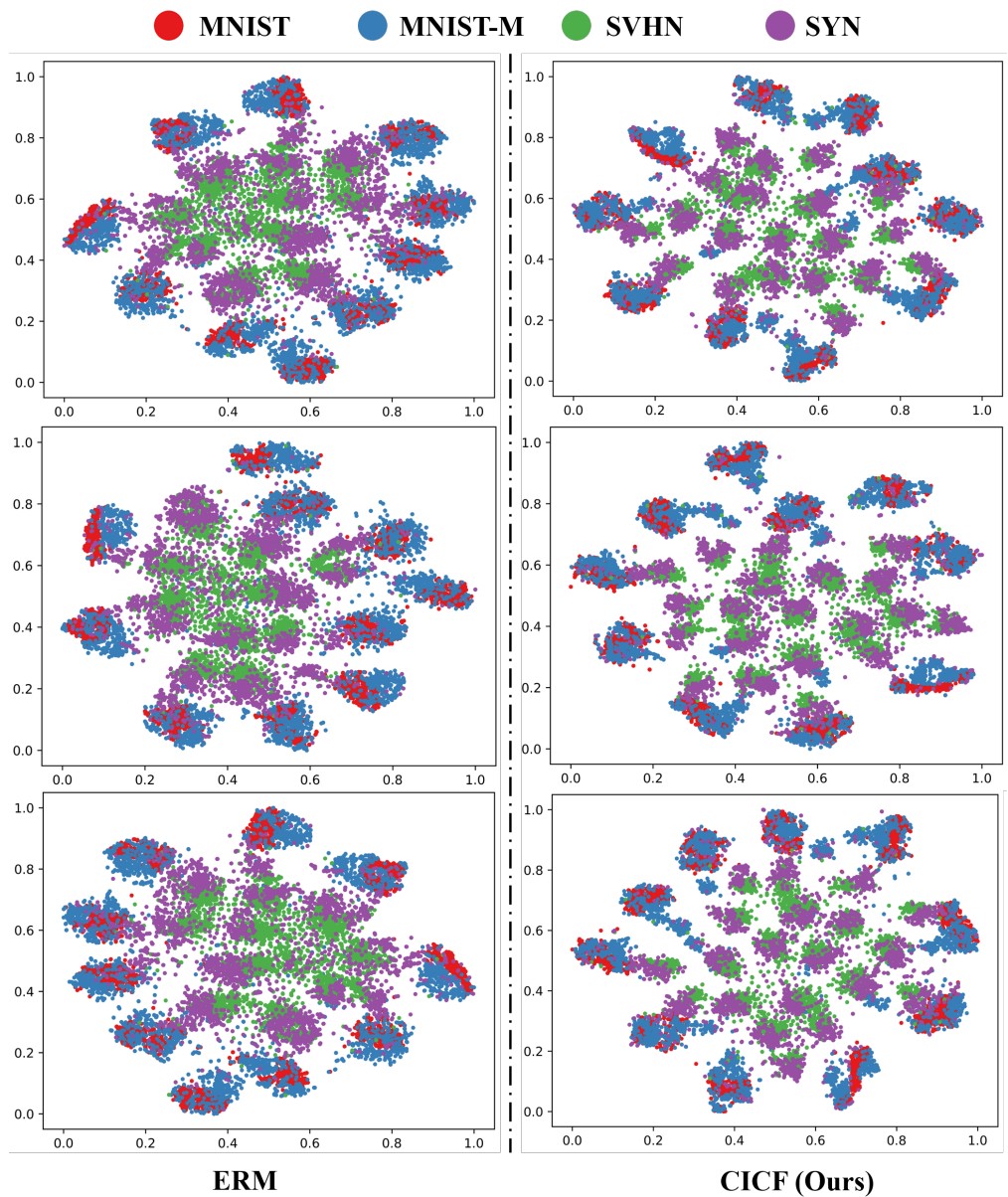

Figure 10: t-SNE visualization of features learned by ERM (left) and our *CICF* (right) with different seeds.

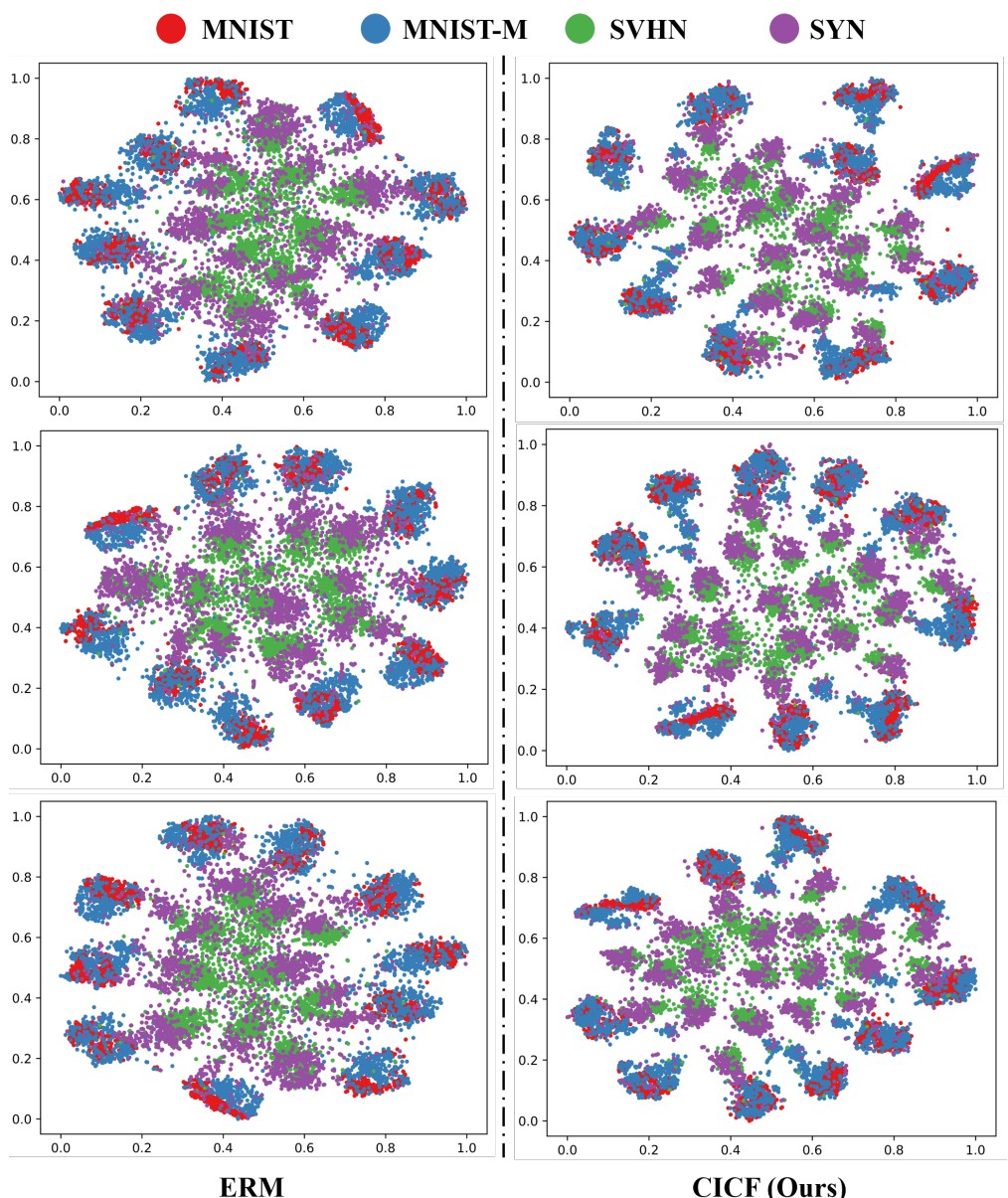

Figure 11: t-SNE visualization of features learned by ERM (left) and our *CICF* (right) with different seeds.

