# OpenReview forum: "Confounder Identification-free Causal Visual Feature Learning"
_ICLR.cc/2023/Conference — Submitted to ICLR 2023_

### Official Review · Reviewer_K8gB · 2022-10-21

**Confidence:** 5
**Correctness:** 1
**Technical Novelty And Significance:** 1
**Empirical Novelty And Significance:** Not applicable
**Recommendation:** 1

**Clarity, Quality, Novelty And Reproducibility:**

This paper has very serious flaws; I didn't check reproducibility because I couldn't get past all the errors in the method. The writeup could also use a serious proof read - there are too many typos and ambiguous statements to list here.

**Strength And Weaknesses:**

This papers is built on a series of serious misunderstandings of the relationship between a model and a data generating process in causal inference. The front door criterion says that if we have access to data of the form X -> Z -> Y, where Z mediates the relationship between X and Y, then we can control for any confounding between X and Y (including unobserved confounding) by the front door adjustment formula. But not - this is a statement about the true data generating process --- i.e. Z is really some variable in the real world that we observe --- it is not a statement about our model of the data generating process, as this paper assumes. Their key idea is to treat the hidden representation as a mediator, which they attempt to justify in appendix A.3, but this justification is just wrong: the hidden representation z = g(X) is *not* on the **causal** path from X to Y. It is just an intermediate step in the computation of model's estimate,  \hat{Y}. We can see this by noting that if we were to intervene on z = g(X) (e.g. by setting it to zero), nothing would change in the true Y (because our model's estimate is not on the causal path of the true data generating process!), but \hat{Y} would change.

Similarly, the section on the gradient-based instantiation of FEIE seriously misunderstanding what an interventional distribution is (again, it's a property of the true data generating process, not the model), so nothing that's written there makes sense from a causal perspective.

**Summary Of The Paper:**

This paper aims to address the fact that deep networks may condition on spurious features that are correlated with the target class without being causal. They attempt to leverage the front door criterion in order to avoid the need to explicitly condition on confounders. Their approach works by treating the hidden representation as a mediator (this is incorrect) and then attempting to control for confound via a front door-like update (except they represent interventions as MAML-style gradient perturbed parameters).

**Summary Of The Review:**

This paper has very serious technical flaws and is not ready for publication/

---

> ### Author Response · Authors · 2022-11-18
> **Response to Reviewer K8gB**
>
> Thanks for your efforts in reviewing our paper. We clarify the reasonableness of our proposed modeling below.
>
> We think your concern lies in that whether the front-door criterion can be used in model learning and whether the front-door criterion is limited to only describing the real data generation process as your thought.
>
> 1) The answer is that the front-door criterion can be used in both the model learning process and the real data generation process. In the real data generation process, the front-door criterion can describe the dataset labeling process by humans, where the mediator is the attributes of the class in the real world. In model learning, the front-door criterion can describe the causal relationship in the model optimization process, where $X$ are the images, $Z$ are the hidden representations, and $Y$ are the predicted labels. We expect the predicted labels $Y$ to be consistent with the ground-truth labels in the datasets by optimization. In this process, selecting the hidden representation of the model as the mediator is reasonable for our front-door modeling, which is the cause of the model prediction $Y$. The reasonability of our front-door criterion in model learning is also proved in [1]. This work also models the front-door criterion for the model learning process and selects the hidden representation as the mediator for the vision-language task. Depart from that, selecting the hidden representations of models as variables to construct the structural causal model is popular in causal inference-based model learning [2, 3, 4, 5, 6, 7, 8, 9, 10, 11]. For instance, IFST [2] establishes the causal graph with the features of the model, pretrained knowledge, and the classification label. [3] utilize latent vectors in the model to represent the semantics of one image from the structural causal graph. VC-RCNN [4]  models the causal graph and the intervention between different visual concepts at the feature space of RCNN.
>
> 2) An intervention should be reasonable and satisfy the real data for the front-door criterion in both model learning and the real data generation process. For instance, in the real data generation process, it is not possible to set the mediator corresponding to a "dog" by the value of the mediator of a "cat", and claim its true label is invariant and is still as a "dog". This is clearly unreasonable. With the same principle, it is also not reasonable to set $z=h(x)=0$ for the model learning as an intervention, since no corresponding $x$ exists in the real data that can let $z=h(x)=0$, which is improper. Furthermore, when the reasonable interventions (such as changing the domain) from the data were added to $z=h(x)$, the predicted labels $Y$ of our model are expected to be invariant and consistent with the groud-truth labels provided by the dataset, which is the optimization direction of our CICF.
>
> [1] Xu Yang, Hanwang Zhang, Guojun Qi, and Jianfei Cai. Causal attention for vision-language tasks. (CVPR2021)
>
> [2] Zhongqi Yue, Hanwang Zhang, Qianru Sun, and Xian-Sheng Hua. Interventional few-shot learning. (NeurIPS 2021)
>
> [3] Chang Liu, Xinwei Sun, Jindong Wang, Haoyue Tang, Tao Li, Tao Qin, Wei Chen, and Tie-Yan Liu. Learning causal semantic representation for out-of-distribution prediction. (NeurIPS 2021)
>
> [4] Tan Wang, Jianqiang Huang, Hanwang Zhang, and Qianru Sun. Visual commonsense r-cnn. (CVPR2020)
>
> [5]  Xinting Hu, Kaihua Tang, Chunyan Miao, Xian-Sheng Hua, and Hanwang Zhang. Distilling causal effect of data in class-incremental learning. (CVPR2021)
>
> [6] Dong Zhang, Hanwang Zhang, Jinhui Tang, Xiansheng Hua, and Qianru Sun. Causal intervention for weakly-supervised semantic segmentation. (NeurIPS 2020)
>
> [7] Tan Wang, Chang Zhou, Qianru Sun, and Hanwang Zhang. Causal attention for unbiased visual recognition. (ICCV2021)
>
> [8] Divyat Mahajan, Shruti Tople, and Amit Sharma. Domain generalization using causal matching. (ICML2021)
>
> [9] Xinwei Sun, Botong Wu, Xiangyu Zheng, Chang Liu, Wei Chen, Tao Qin, and Tie-Yan Liu. Recovering latent causal factor for generalization to distributional shifts. (NuerIPS 2021)
>
> [10] Tan Wang, Jianqiang Huang, Hanwang Zhang, and Qianru Sun. Visual commonsense representation learning via causal inference. (CVPRW 2020)
>
> [11] Zhongqi Yue, Qianru Sun, Xian-Sheng Hua, and Hanwang Zhang. Transporting causal mechanisms for unsupervised domain adaptation. (ICCV2021)

---

> > ### Author Response · Authors · 2022-12-02
> > **To Reviewer K8gB**
> >
> > Thank you very much for your review. Welcome to raise new questions if you have.

---

> > > ### Comment · Reviewer_K8gB · 2022-12-07
> > > **Response**
> > >
> > > Sorry about the delay in responding.
> > >
> > > 1) "In model learning, the front-door criterion can describe the causal relationship in the model optimization process, where X are the images, Z are the hidden representations, and Y are the predicted labels. We expect the predicted labels Y to be consistent with the ground-truth labels in the datasets by optimization."
> > >
> > > I don't understand what you mean by this: if it were true that you could correctly adjust for confounding using a hidden representation, then we could also use the front door criterion on classical causal inference tasks (e.g. take any classical problem from Pearl's textbooks). What makes the image setting special that the standard negative results of causal inference don't apply? Causal inference with unobserved confounding is impossible without additional structure such as an instrumental variable or an *observed* mediator (in which case the front door criterion would apply). In this setting you're learning the hidden representation, so it is *not* observed.
> > >
> > > I didn't look at all those references, but you are correct that [1] makes the same mistake as this paper. That doesn't it correct though.
> > >
> > > 2) Sure - you typically need overlap. But that was just a super simple example to make it clear that the model is not on the causal pathway. One could equivalently intervene to swap the hidden representation with any other class's hidden representation, and you'll swap the label of the model's prediction but not the true label in the data.

---

> > > > ### Author Response · Authors · 2022-12-09
> > > > **To Reviewer K8gB**
> > > >
> > > > It is really appreciated your response and review of our paper.  We expect the following responses to be helpful to understand our work and solve your confusion about work.
> > > >
> > > > **Q1:**  I don't understand what you mean by this: if it were true that you could correctly adjust for confounding using a hidden representation, then we could also use the front door criterion on classical causal inference tasks (e.g. take any classical problem from Pearl's textbooks). What makes the image setting special that the standard negative results of causal inference don't apply? Causal inference with unobserved confounding is impossible without additional structure such as an instrumental variable or an observed mediator (in which case the front door criterion would apply). In this setting, you're learning the hidden representation, so it is not observed.
> > > >
> > > > **A1:**  **1)** We model the front-door criterion in the model learning process, which aims to simulate the causal relationship in the real world.  In our paper, we model the intervention effects from all samples on the current sample by considering the influence of those samples on the network parameters in optimization. In this way, we do not need to explicitly identify the confounding factors. If we could model a classical causal inference task as you described as a network fitting problem, we believe our method is applicable to that task.   **2)** The hidden representation $Z$ of the networks can be observed since it is the output of the sub-model $h$. For example, there is a series of works that exploit the observed hidden representation for causal inference[1,2,3,4,5] or other tasks [6,7,8].
> > > >
> > > > [1] Yang X, Zhang H, Cai J. Deconfounded image captioning: A causal retrospect[J]. IEEE Trans-
> > > > actions on Pattern Analysis and Machine Intelligence, 2021 (TPAMI2021).
> > > >
> > > > [2] Yang M, Liu F, Chen Z, et al. CausalVAE: Disentangled representation learning via neural structural causal models. (CVPR2021)
> > > >
> > > > [3] Liu C, Sun X, Wang J, et al. Learning causal semantic representation for out-of-distribution prediction[J]. (NeurIPS2021)
> > > >
> > > > [4] Divyat Mahajan, Shruti Tople, and Amit Sharma. Domain generalization using causal matching. (ICML2021)
> > > >
> > > > [5] Tan Wang, Jianqiang Huang, Hanwang Zhang, and Qianru Sun. Visual commonsense r-cnn. (CVPR2020)
> > > >
> > > > [6] Rombach R, Blattmann A, Lorenz D, et al. High-resolution image synthesis with latent diffusion models (Stable Diffusion CVPR2022)
> > > >
> > > >
> > > >
> > > > [7] Liu H, Song Y, Chen Q. Delving StyleGAN Inversion for Image Editing: A Foundation Latent Space Viewpoint. (ICLR2021)
> > > >
> > > > [8] Esser P, Rombach R, Ommer B. Taming transformers for high-resolution image synthesis. (CVPR2021)
> > > >
> > > >
> > > > **Q2:** Sure - you typically need overlap. But that was just a super simple example to make it clear that the model is not on the causal pathway. One could equivalently intervene to swap the hidden representation with any other class's hidden representation, and you'll swap the label of the model's prediction but not the true label in the data.
> > > >
> > > > **A2:** Actually, our causal modeling is established on the model optimization process, and the sub-model $h$ and $f$ are on the causal pathway of model optimization. The purpose is to reduce the confounding effects for the generalization of models caused by the confounders that existed in the datasets. In our model training process, we do not perform/add the intervention by swapping/changing the hidden representation of different classes. We model the intervention from all samples on the current sample, which is reflected in the optimization of the network parameter $y=f(z)$. From another perspective, once the causal relationship $P(Y|do(X))$ is learned, the testing process only requires setting the $X=x$, instead of setting a $Z$ as a specific value. Moreover, We believe that the Case 4.2.3 in [9] is helpful to understand the difference between the testing process by the estimated casual model, and the training process to get a casual model by casual inference with your said data.
> > > >
> > > > [9] Pearl J, Glymour M, Jewell N P. Causal inference in statistics: A primer. 2016[J].
> > > >
> > > > Feel free to raise questions if you have any.

---

### Official Review · Reviewer_pVYZ · 2022-10-24

**Confidence:** 4
**Correctness:** 3
**Technical Novelty And Significance:** 3
**Empirical Novelty And Significance:** 3
**Recommendation:** 5

**Clarity, Quality, Novelty And Reproducibility:**

Clarity:   As stated in the question, some parts of the paper is not well explained.

Quality: OK.

Novelty: Good

Reproducibility: Not known.

**Strength And Weaknesses:**

Strength:
I like the way that the author try to use different subfield to address the domain generalization problem with meta learning and causal inference.

Weakness:
Please see the questions:



Question:
The original front door criteria is actually a concatenation of two time back-door criterion, which is concatenating two times do operation. The first do operation is p(Z=z|do(X=x)), which means if one cut the parent of node X, what would be the conditional distribution of Pr_{modified graph}(Z=z|X=x), usually, this z is observed and believed to be a direct cause of target variable Y and x is a direct cause of z.


Why p(Z=z|X=x)= 1? If this is true, then $z$ should be a deterministic map of X.

For the second backdoor part $\sum_x Pr(Y|Z, x) p(x)$, which is essentially treating x as an indirect confounder, due to the deterministic map z=h(x), the summation over z is not needed, Pr(Y|Z, X) only depend on X. As the author has observed, directly concatenating $z=h(x)$ would result in $z$ being ignored. But the author introduced using a “look-ahead” (name I give) gradient to represent this probability, i did not see much explanation why?


**Summary Of The Paper:**

To eliminate domain effect in deep learning, existing approach relies on identifying explicitly the confounder and remove confounder effect via backdoor criterion.  The proposed method implements front door criterion via gradient.

The author proposed to use a neural network to represent $\sum_{x}Pr(Y|Z=z, x)$, and shed light on the connection with Model Agnostic Meta Learning, which is, similar to MAML, use a “look-ahead” gradient operation to represent the conditional distribution.

To achieve this look ahead, the author proposed to sample a small portion of the whole sample which represent the whole distribution.
To achieve a sampling strategy which could mimic the all-observation distribution, the author use K-means algorithm to cluster the data into several clusters and take a small portion from each of the cluster.


**Summary Of The Review:**

I think the paper is very innovative but still lacks some detailed explanation to the crucial part. Especially the way that the author use look-ahead gradient step to approximate the conditional distribution and using a neural network to extract a deterministic feature from image X to z, which is not how front door criterior works initially.

---

> ### Author Response · Authors · 2022-11-18
> **Response to Reviewer pVYZ**
>
> **Q1**: The original front door criteria is actually a concatenation of two time back-door criterion, which is concatenating two times do operation. The first do operation is $P(Z=z|do(X=x))$, which means if one cut the parent of node X, what would be the conditional distribution of $P(Z=z|X=x)$, usually, this $z$ is observed and believed to be a direct cause of target variable $Y$ and $x$ is a direct cause of $z$. Why $p(Z=z|X=x)= 1$? If this is true, then  $z$ should be a deterministic map of $X$.
>
> **A1**: Thanks for your comments. As described in Appendix 3, we have provided the explanation/proof for why utilizing $Z=h(x)$ in our paper satisfies the front-door criterion.   Now, let me introduce why we select the Z as the middle feature and let $P(Z=z|X=x)=1$.  From the causality, it is essential that Z and X do not have common confounders, and that $Z$ blocks all the paths from X to Y. However, it is not easy to find a real vector in the real world to satisfy that. The middle feature, as a special middle vector that satisfies the above principles, is easy to observe and used in the training of the networks, which is also used in [1]. The middle feature Z is decided by two factors, the input X and model parameters $h$ as $Z = h(X)$. When we observe the middle feature $Z$ at the inference stage, the model parameters h are fixed. And Z is only determined by $X$. It means if we gave $X=x$, only one $Z=z$ can be got from $z=h(x)$. It means $Z$ should be a deterministic map of $X$. Therefore, we set $p(Z=z|X=x)=1$, which simplifies the implementation of the front-door criterion in causal learning.
>
> **Q2**: For the second backdoor part, which is essentially treating x as an indirect confounder, due to the deterministic map z=h(x), the summation over z is not needed, P(Y|Z, X) only depends on X. As the author has observed, directly concatenating would result in being ignored. But the author introduced using a “look-ahead” (name I give) gradient to represent this probability, I did not see much explanation why?
>
> **A2**: Thanks for your comments. From the $P(Y|Z,X)$, we can find that this equation only provides the theoretical inspiration for us that we have to model the intervention from $X$ to $P(Y|Z)$. But it does not provide an intervention modeling strategy indeed. One simple solution is to directly concatenate them in the feature space. But it is implicit and requires some design to let $X$ produce the intervention for $Z$. Otherwise, we cannot ensure whether the model can extract the intervention properly for $Z$, or even ignore the intervention. Different from the above strategy, we propose modeling the intervention from $X$ to $Z$ by a "look-ahead" gradient. The reasons are as: 1) The classification network learns the probability of $P(Y|Z)$ from end-to-end optimization, which is based on the gradient from each $X=x$. When the model is trained with the sample $X=x$, the parameter is updated with "look-ahead", which is conditioned on the samples $X$. Therefore, when based on the updated parameters, when we maximize the $P(Y|Z)$, it is equivalent to maximizing $P(Y|Z, X)$, since the model has been conditioned on sample $X$ through the parameters. Then the optimization direction related to $Z$ for the model parameters will confront the intervention from $X$ while maximizing the $P(Y|Z, X)$, and cut off the connection from $X$ to Z at the second back-door criterion.

---

> > ### Author Response · Authors · 2022-12-02
> > **To Reviewer pVYZ**
> >
> > Thank you very much for your review. It is appreciated for that if you have other suggestions. Welcome to raise new questions if you have.

---

> > ### Comment · Reviewer_pVYZ · 2022-12-08
> > **response to A1:**
> >
> > Thanks for the author to offer response to my Q1 in A1:
> > I have difficulty to validate the proof in your appendix:
> > Quote:
> > "As shown in Fig. 2, the direct path from X to Y is built from the model φ, that is composed
> > of two sub-models, h and f. We can represent this path as Z = g(X), Y = f(Z). We can
> > observe that Z intercepts all directed paths from X to Y , which satisfies the Principle 1)"
> >
> > Actually, the requirement for a front door variable should be
> > " 1) Z intercepts all directed paths from X to Y"
> >
> > This requirement is a joint distribution requirements for random variables X, Z, Y, how simply training a neural network could ensure this joint distribution requirements?

---

> > > ### Author Response · Authors · 2022-12-09
> > > **To Reviewer pVYZ**
> > >
> > > Thanks for the author to offer a response to my **Q1** in **A1**. We believe the following response can solve your concerns.
> > >
> > > **Q1:** I have difficulty to validate the proof in your appendix: Quote: "As shown in Fig. 2, the direct path from $X$ to $Y$ is built from the model φ, that is composed of two sub-models, $h$ and $f$. We can represent this path as $Z = g(X)$, $Y = f(Z)$. We can observe that $Z$ intercepts all directed paths from $X$ to $Y$, which satisfies the Principle 1)"
> > >
> > > Actually, the requirement for a front door variable should be " 1) $Z$ intercepts all directed paths from $X$ to $Y$"
> > >
> > > This requirement is a joint distribution requirement for random variables $X$, $Z$, $Y$, how does simply training a neural network could ensure this joint distribution requirement?
> > >
> > > **A1:** Thank you for the question.  Actually, in the early training stage, the sub-networks $ f$  and $h$ are jointly trained with ground-truth label supervision, which enables $Z=h(X)$ to reasonably intercept the paths from $X$ to $Y$, avoiding the loss of necessary information in $Z$. The variables $X$, $Z$, and $Y$ satisfy the joint distribution requirements. In addition, this front-door modeling scheme with two sub-models and applying the middle representation as $Z$ also can be found in the [1, 2] but the intervention scheme is not the same as ours.
> > > We will add more details in the approach section to clarify this.
> > >
> > > [1] Yang X, Zhang H, Cai J. Deconfounded image captioning: A causal retrospect[J]. IEEE Transactions on Pattern Analysis and Machine Intelligence, 2021 (TPAMI2021).
> > >
> > > [2] Yang X, Zhang H, Qi G, et al. Causal attention for vision-language tasks[C]//Proceedings of the IEEE/CVF Conference on Computer Vision and Pattern Recognition. 2021: 9847-9857.
> > >
> > >
> > > Feel free to raise questions if you have any.

---

### Official Review · Reviewer_wmrP · 2022-10-24

**Confidence:** 3
**Correctness:** 3
**Technical Novelty And Significance:** 4
**Empirical Novelty And Significance:** 2
**Recommendation:** 5

**Clarity, Quality, Novelty And Reproducibility:**

The paper is novel and clearly written. The authors provide training details in the appendix, but the code is not provided as far as I understood.

**Strength And Weaknesses:**

**Strengths:**

The paper introduces a novel idea to use the front-door criterion instead of the back-door for covariate adjustment. They theoretically demonstrate how to incorporate front-door criteria ideas into the learning algorithm. They additionally demonstrate that front-door criteria could be used to theoretically explain the previously proposed meta-learning algorithm (MAML).

**Weaknesses and questions:**

- Why the approach proposed by eq. (4) won’t also lead to a trivial solution? Won’t the gradient become close to zero when the model converges? Won’t be the information on x tilda also ignored?
- It is not clear to me why results for previously published models don’t agree with the DomainBed framework. The experiments would look more convincing if they were implemented and compared within DomainBed framework and good reproducibility practices.
- Grad-CAM maps from supplementary figure 9 look less optimistic than the ones in figure 4. For example, it shows that the model still attention to the bone to classify the dog. Also, such visualizations would be more interesting to compare not to models learned with ERM strategy, but with any causal learning method, e.g. IRM.
- Also for synthetic examples in order to show the effect of the confounding variable was removed, it would be more interesting to see results on colored MNIST.
- A more explicit demonstration of the removed confounding effect would be very helpful.
- The results in Table 1,3,4 are better on average, but not across all environments. If the model was truly correcting confounding, why would be there such inconsistencies?
- Figure 9 is also not very convincing that the learned features are somehow significantly different. The plots look actually quite similar up to random seed or some t-SNE hyperparameters.
- Confidence intervals in Tables 1,3,4 would be quite helpful.
- How did the authors choose the best set of hyper-parameters for each dataset?


**Summary Of The Paper:**

In this paper, the authors propose a novel method (CICF) that is based on the idea of front-door adjustment for hidden confounders. They additionally demonstrate the relation between CICF and the popular meta-learning strategy MAML (Finn et al., 2017). The authors provide a theoretical interpretation of why MAML works.

**Summary Of The Review:**

I find the proposed method and theoretical part of the part quite interesting, but I am not convinced by the experiments (described above my questions and concerns).

---

> ### Author Response · Authors · 2022-11-18
> **Response to Reviewer wmrP**
>
> **Q1**: Why the approach proposed by Eq. (4) won’t also lead to a trivial solution? Won’t the gradient become close to zero when the model converges? Won’t be the information on $\tilde{x}$ also ignored?
>
> **A1**: Let us analyze the condition for reaching the trivial solution. The condition is “All gradients of $\tilde{x}$ must be close to zero in the whole training process (from start to end) instead of only the converging state (i.e., the end of training)". When the above condition satisfies, the network will be hard to capture the interventional effects from all other samples in the whole training process. But, in our practical training process, supervised by classification loss, it is impossible for the randomly initialized (or ImageNet pretrained) network to have the gradients for all $\tilde{x}$ to be close to 0 before it converges. During the training (especially before the convergence), the information on $\tilde{x}$ is leveraged to learn to eliminate the intervention from other samples.
>
> **Q2**: It is not clear to me why results for previously published models don’t agree with the DomainBed framework. The experiments would look more convincing if they were implemented and compared within DomainBed framework and good reproducibility practices.
>
> **A2**: Thanks for your suggestions about benchmarks.  ***It is noteworthy that there are two differences between our experimental settings with the DomainBed[1], which cause this disagreement***. 1) The backbone used in DomainBed[1] is ResNet-50 for PACS, VLCS, and OfficeHome. In Tables 1, 3, and 4, we follow previous works[2,3,4,5,6] and use the commonly-used backbones ResNet-18 for PACS, Office-Home, and AlexNet for VLCS, which is different from DomainBed[1]. 2) For MINIST, the used datasets are different. Following MixStyle[2] and FACT[3], we use Digits-DG as the dataset instead of the Color MINIST used in DomainBed[1] Benchmarks.  ***Following your suggestion, we have implemented and compared it within DomainBed[1] framework. We follow the DomainBed and conduct the experiments on VLCS, Office-Home with ResNet-50, and Color MINIST with the MNIST backbone in DomainBed[1] and show the detailed results in the Appendix A.7 of our revision, including Table 5, 6 and 7.***  Our
> CICF improves the ERM by an average gain of 2% on Office-Home, 1.7% on VLCS and 4.1% on ColorMNIST. Moreover, our CICF achieves the best performances on these three datasets.
> It is noteworthy that even for the most difficult domain "0.9" in ColorMNIST, our CICF can achieve an accuracy of 21.4%, which exceeds the second method MMD (Li et al., 2018b) by 10.9%
> |  Algorithm  |  Office-Home | VLCS | ColorMNIST|
> |  ----  | ----  | ----  | ----  |
> | ERM | 67.5 ± 0.5 | 77.4 ± 0.3 | 52.0 ± 0.1|
> |ERM+CICF(Ours)| 69.5 ± 0.2 | 79.1 ± 0.3 | 56.1 ± 0.2|
> |
>
>
>
> [1] Ishaan Gulrajani and David Lopez-Paz. In search of lost domain generalization. (ICLR 2021)
>
> [2] Kaiyang Zhou, Yongxin Yang, Yu Qiao, and Tao Xiang. Domain generalization with mixstyle. (ICLR 2021)
>
> [3] Qinwei Xu, Ruipeng Zhang, Ya Zhang, Yanfeng Wang, and Qi Tian. A fourier-based framework for domain generalization. (CVPR 2021)
>
> [4] Da Li, Yongxin Yang, Yi-Zhe Song, and Timothy M Hospedales. Learning to generalize: Meta-learning for domain generalization. (AAAI 2018)
>
> [5] Yogesh Balaji, Swami Sankaranarayanan, and Rama Chellappa. Metareg: Towards domain generalization using meta-regularization. (NeurIPS 2018)
>
> [6] Qi Dou, Daniel Coelho de Castro, Konstantinos Kamnitsas, and Ben Glocker. Domain generalization via model-agnostic learning of semantic feature. (NeurIPS 2019)
>
> **Q3**: Grad-CAM maps from supplementary figure 9 look less optimistic than the ones in figure 4. For example, it shows that the model still pays attention to the bone to classify the dog. Also, such visualizations would be more interesting to compare not to models learned with ERM strategy, but with any causal learning method, e.g. IRM.
>
> **A3**: Thanks for the suggestion. ***We have added the visualization of IRM in our revision.***  For the image of the dog with a bone in the Grad-cam of Fig. 8 (not Fig. 9),  we can see that ours also focuses on the head of a dog, but ERM only focuses on the bone, which reveals the effectiveness of our methods.  Note that the Grad-CAM maps in Fig. 4 and Fig. 9 (Fig. 8) are obtained from different target domains from PACS, i.e., sketch/painting in Fig. 4, and cartoon/photo in Fig.8, respectively. ***For different target domains (unseen in training), the training domains are composed of the other three domains, where the training domains and their confounders are also different. Therefore, the performance is also different.***

---

> > ### Author Response · Authors · 2022-11-18
> > **Response to Reviewer wmrP-Part 2**
> >
> > **Q4**: Also for synthetic examples in order to show the effect of the confounding variable was removed, it would be more interesting to see results on colored MNIST.
> >
> > **A4**: Following your suggestion, we have conducted experiments on colored MNIST. We follow the DomainBed[1] and do not use any data augmentation. The experimental results show that our strategy brings significant gain, i.e., 4\%, over the baseline ERM.  The detailed results and comparisons are listed in Table. 7 of Appendix.
> > | Algorithm | 0.1 | 0.2 | 0.9 | Avg. |
> > |----|----|----|----|----|
> > |ERM| 72.7 ± 0.2 | 73.2 ± 0.3 | 10.0 ± 0.0 | 52.0 ± 0.1|
> > |ERM+CICF(Ours) | 72.9 ± 0.3 | 74.1 ± 0.2 | 21.4 ± 0.4| 56.1 ± 0.2|
> > |
> >
> >
> > **Q5**:  A more explicit demonstration of the removed confounding effect would be very helpful.
> >
> > **A5**: Thanks for your suggestion and we agree with this. However, in the real world, confounders are hard to identify. We will attempt to explore how to provide an explicit demonstration in our future work.
> >
> > **Q6**: The results in Table 1,3,4 are better on average, but not across all environments. If the model was truly correcting confounding, why would there be such inconsistencies?
> >
> > **A6**: Thanks for the insightful question. This may be because the confounding factors in the testing data are not always the same as that in the training data. Therefore, the benefits of eliminating the confounding factors based on training data are not consistent in all environments.
> >
> > **Q7**: Figure 9 is also not very convincing that the learned features are somehow significantly different. The plots look actually quite similar up to random seed or some t-SNE hyperparameters.
> >
> > **A7**: We utilize the default hyperparameters in the t-SNE tools. We have observed that the differences between different random seeds are smaller than the differences between the two models. We provide a visualization of different seeds in t-SNE in Fig. 10 and Fig. 11 of the Appendix. The conclusion is not dependent to the seeds in t-SNE tools.
> >
> > **Q8**: Confidence intervals in Tables 1,3,4 would be quite helpful.
> >
> > **A8**: Following your nice suggestion, we have added the confidence intervals in Tables 5,6, and 7 on DomainBed.  Limited to the paper formatting, we will further add the confidence intervals for Tables 1, 3, and 4 in the final version.
> >
> > **Q9**: How did the authors choose the best set of hyper-parameters for each dataset?
> >
> > **A9**: For different datasets, we only need to choose the hyper-parameter $\alpha$ (see Eq. (4)) which denotes the learning rate for the virtual update. We determine it by observing the loss functions. If the loss functions become too larger, the learning rate is not proper. We will change the interval of the learning rate. Our method is not very sensitive to hyper-parameter. We set it to 0.05, 0.005, 0.5 for PACS, VLCS, and Digit-DG, respectively.

---

> > > ### Author Response · Authors · 2022-12-02
> > > **To Reviewer wmrP**
> > >
> > > Thank you very much for your review. Welcome to raise new questions if you have.

---

### Official Review · Reviewer_jBAD · 2022-10-24

**Confidence:** 3
**Correctness:** 4
**Technical Novelty And Significance:** 3
**Empirical Novelty And Significance:** 3
**Recommendation:** 8

**Clarity, Quality, Novelty And Reproducibility:**

The approach is presented clearly, as well as the various learning strategies involved. There is a thorough description of the algorithm as well as various choices for subnetworks and feature visualization as well.

**Strength And Weaknesses:**

Strength: Uses a front-door graph instead of a back door graph, which has novelty. Describes how to fit intractable models and achieves state of the art results on  datasets.

Weakness: A big challenge with causal inference is being able to correctly fit the nuisance models when we are interested in learning the target parameter. Here, I’d like to see some more assurance around the proposed training strategy being able to fit the nuisance models correctly and consequently correct learn the interventional distributions in a variety of situations.

**Summary Of The Paper:**

This paper presents a novel method of learning robust DNNs through the use of a network containing two si models, one that learns an intermediate variable Z and then combines it with the front door criterion to learn the interventional distribution p(Y|do(X = x)). This interventional distribution has invariance properties, and that is why it is a common learning strategy used when it comes to building robust models. Various strategies to learn certain intractable models are described, and state of the art performance is shown on certain datasets. Additionally, this provides a theoretical justification for an existing approach, which is appreciated.

**Summary Of The Review:**

This paper uses a two stage network to induce a front door causal graph and learn the interventional distribution p(Y | do(X+x)). since this interventional distribution is invariant, it is robust to distribution shift and domain generalization. This allows the authors to learn a prediction model that is robust and outperforms state of the art methods.

---

> ### Author Response · Authors · 2022-11-18
> **Response to Reviewer jBAD**
>
> **Q1**: A big challenge with causal inference is being able to correctly fit the nuisance models when we are interested in learning the target parameter. Here, I’d like to see some more assurance around the proposed training strategy being able to fit the nuisance models correctly and consequently correctly learn the interventional distributions in a variety of situations.
>
> **A1**: Thanks for your positive comments and thoughtful review. Actually, for datasets with different characteristics, our training strategy consistently achieves significant performance improvements over the baseline models. For example, PACS contains images from four domains, i.e., Photo, Art painting, Cartoon, and Sketch, each with a different style and distribution. From the visualization results (Figure 4), we can see that the baseline model usually focuses on parts of the related objects while our model is prone to focus on the entire objects, thanks to the capability of eliminating the confounding effects to learn generalizable representations. We will take your suggestions and explore the effectiveness of the training strategy in more real-world scenarios in the future.

---

### Author Response · Authors · 2022-11-18
**To All reviewers**

We thank all reviewers and area chairs for their great efforts!  We thank the first three reviewers for the recognition of the novelty of our work and the constructive suggestions/comments. The fourth reviewer may have some misunderstanding of our work and we will try to clarify the perspective. We address the questions/concerns in detail. Feel free to raise questions if you have any.

---

### Decision · Program_Chairs · 2023-01-20

**Decision:**

Reject

**Justification For Why Not Higher Score:**

this paper has technical flaws

**Justification For Why Not Lower Score:**

n/a

**Metareview: Summary, Strengths And Weaknesses:**

This paper proposes a visual feature learning method that learns a latent variable between x and y and then apply front-door criterion to adjust for confounding effects. The relation between the proposed method and the meta learning method MAML is also discussed. The proposed method is applied to domain generalization.

As pointed out by one of the reviewers, this paper makes a fundamental mistake. The front door criterion is based on an observable mediation variable that adjusts for the effect of hidden confounders. However, the proposed method assumes that the network learns an intermediate latent variable which can serve for the purpose of de-confounding, which is basically wrong. One cannot conditioning on a latent variable in the front door adjustment.  Thus, I would recommend rejection of this paper.